# Temporal Difference Calibration in Sequential Tasks: Application to Vision-Language-Action Models

**Shelly Francis-Meretzki** [* 1]  **Mirco Mutti** [1]  **Yaniv Romano** [1]  **Aviv Tamar** [1]

https://shellytechnion.github.io/TDQC.github.io

## Abstract

Recent advances in vision-language-action (VLA) models for robotics have highlighted the importance of reliable uncertainty quantification in sequential tasks. However, assessing and improving calibration in such settings remains mostly unexplored, especially when only partial trajectories are observed. In this work, we formulate *sequential calibration* for episodic tasks, where task-success confidence is produced along an episode, while success is determined at the end of it. We introduce a sequential extension of the Brier score and show that, for binary outcomes, its risk minimizer coincides with the VLA policy's value function. This connection bridges uncertainty calibration and reinforcement learning, enabling the use of temporal-difference (TD) value estimation as a principled calibration mechanism over time. We empirically show that TD calibration improves performance relative to the state-of-the-art on simulated and real-robot data. Interestingly, we show that when calibrated using TD, the VLA's single-step action probabilities can yield competitive uncertainty estimates, in contrast to recent findings that employed different calibration techniques.

## 1. Introduction

Modern neural network based AI systems are typically "black box", raising concern in applications that require safety and reliability – where a system must not only be accurate but also *correctly assess when it is likely to fail*.

While neural networks often provide a confidence together with their prediction, there is no built-in machinery to enforce that the confidence actually matches the probability of being correct. In this context, *calibration* explicitly requires that a model's confidence aligns with the empirical outcome frequencies (Guo et al., 2017; Brier, 1950; Xiong et al., 2023; Jiang et al., 2021). When a model is well calibrated, high confidence predictions are correct most of the time, and low confidence predictions correspond to a higher chance of error. This is a desirable property as it enables downstream safety mechanisms that depend on the model's confidence.

Although calibration has been broadly studied, most of the research focuses on single-step decision problems, such as classification and regression (Popordanoska et al., 2024; Dheur & Ben Taieb, 2023). In these problems, the evaluation of the model's error is direct: Each input is paired with a ground truth label, so that calibration can compare the model's confidence to the empirical frequency of successes. However, neural networks are commonly used in sequential decision problems as well, for example, in reinforcement and imitation learning (Black et al., 2024; Guo et al., 2025). In sequential tasks, it is not obvious how to define success for individual steps. Think of a robot performing a manipulation task, which requires a sequence of inter-dependent "correct" actions to be solved, and success is only assessable at the end of the sequence. This raises a question:

*How to reason about calibration in sequential tasks?*

Differently from previous settings, (i) the ground-truth label is typically delayed; and (ii) the confidence in the current decision may depend on the confidence of future and past decisions.

In this paper, we provide a formal framework for sequential calibration that encompasses (i, ii). We cast the calibration problem as learning a predictor that takes as input the model's confidence over time to minimize a sequential version of the Brier score (Brier, 1950). The Brier score is a popular loss that captures *both the accuracy and calibration* of a predictor by computing the mean squared error between the prediction and the outcome frequency.

Our insight shows that, when formulating a sequential task as a partially-observable Markov decision process, we can connect the problem of minimizing the sequential version of

---

[1]Technion - Israel Institute of Technology. Correspondence to: Shelly Francis-Meretzki <shellyfra@campus.technion.ac.il>.

*Proceedings of the 43rd International Conference on Machine Learning*, Seoul, South Korea. PMLR 306, 2026. Copyright 2026 by the author(s).

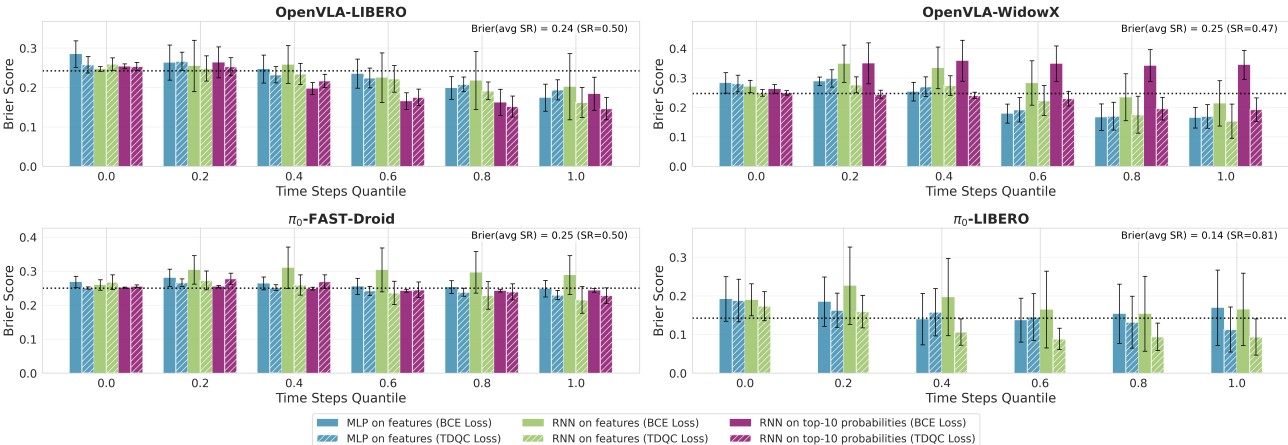

*Figure 1.* **Sequential Brier scores across benchmarks.** Sequential Brier score ($\downarrow$) on an *unseen* validation set averaged over 21 random seeds (train/validation task splits). To compare calibration across rollouts with different lengths, we report Brier score over *time quantiles*. Each subplot corresponds to a (VLA model, benchmark) pair. Success prediction methods are based on sequences of features or action probabilities. Across all settings, our TD-based methods consistently outperform predictors trained with binary cross entropy (BCE). For $\pi_0$ action probabilities are not directly interpretable, hence probability-based TDQC variants are not reported. The dotted line represents the Brier score of a constant predictor that consistently outputs the empirical mean success rate computed on seen tasks (see Remark 4.2).

Brier score to learning a value function, a popular predictor of task success in Reinforcement Learning (RL, Mannor et al., 2026). This fundamental link allows us to bridge algorithmic insights from RL to sequential calibration, such as the ubiquitous Temporal-Difference (TD) loss (Sutton & Barto, 2018; Mnih et al., 2013; Haarnoja et al., 2018), which bootstraps future value predictions to previous steps, resulting in a superior bias-variance tradeoff (Kearns & Singh, 2000). This provides the backbone for our **T**emporal-**D**ifference **Q**-based **C**alibration (TDQC) method.

To showcase the potential of sequential calibration, we turn our attention to Vision-Language-Action (VLA) models. VLAs are powerful architectures that have risen to the state-of-the-art for learning robot policies (Black et al., 2024). VLAs encode decision policies that map visual observations and language instructions to probabilities over control actions. Recent works have targeted calibration of those action probabilities to the long-term task success probabilities (Zollo & Zemel, 2025). In particular, Gu et al. (2025) propose *SAFE*, a method to learn a failure detector that takes as input the hidden state of the VLA and minimizes the cross entropy between the prediction and long-term task success, achieving state-of-the-art failure detection for VLAs. Our paper further advances the research in crucial areas:

- We provide the first formulation for calibration in sequential tasks, unifying recent works on failure detection in VLAs (Zollo & Zemel, 2025; Gu et al., 2025) with the broad calibration literature into a coherent framework that singles out the challenges of the sequential setting;
- We fundamentally link sequential calibration to value prediction in RL, drawing inspiration from the latter to

derive a novel TD method for calibration (TDQC);
- We empirically show that TDQC, accessing only action probabilities, matches or improves the calibration performance of SAFE accessing the hidden state of the policy in challenging sequential tasks (see Fig. 1 top). This is essential to measure calibration of black-box models, for which the hidden state is often not accessible from APIs;
- We show that the TDQC loss, either applied to SAFE or to action probabilities, is an essential ingredient to achieve state-of-the-art early detection results in LIBERO for OpenVLA (Kim et al., 2024), $\pi_0$ (Black et al., 2024), $\pi_0$-FAST (Pertsch et al., 2025) and UniVLA (Wang et al., 2025) models and in a Franka real-robot dataset collected with $\pi_0$-FAST (see Table 2).
- Finally, as a by-product of the TDQC method, we show that the value predictor can improve the baseline's policy success rates by using it to guide the action selection within a set of actions sampled from the policy. In our experiments, this simple recipe yields a 15% increase of the success rate for OpenVLA on LIBERO-10 (see Fig.4).

## 2. Background

### 2.1. Calibration and Accuracy of Predictors

The accuracy of a prediction model is its frequency of making correct predictions. Calibration measures how well a model's predicted confidence aligns with empirical outcome frequencies. We formally discuss these concepts in the context of binary classification.

**Definition 1** (Binary Classification)**.** Let $(X, Y)$ be a pair of random variables where $X \in \mathcal{X}$ denotes an input, $Y \in \{0, 1\}$ is a binary label, and $\mathbb{P}(X, Y)$ is their joint distribu-

tion. A probabilistic classifier is a function $f : \mathcal{X} \to [0, 1]$. For input $x$, the model prediction $f(x)$ is interpreted as the probability that the label is 1.

Typically, $f$ is calculated from an i.i.d. dataset $\mathcal{D} = \{(x_i, y_i)\}_{i=1}^N \sim \mathcal{P}^N$, where $(x_i, y_i)$ are drawn from $\mathbb{P}(X, Y)$. The Brier score (Brier, 1950) is a strictly proper scoring rule that measures the accuracy of a predictor.

**Definition 2** (Brier Score for Binary Classifier)**.** The Brier score is the mean squared $\ell_2$ distance between the prediction and the ground-truth label $\mathrm{BS}(f) = \mathbb{E}_{X,Y}\left[(f(X) - Y)^2\right]$. Given a dataset $\mathcal{D}$, it can be estimated as $\widehat{\mathrm{BS}}(f) = \frac{1}{N}\sum_{i=1}^N (f(x_i) - y_i)^2$.

The Brier score relates to the calibration and accuracy of the classifier by its two-component decomposition. Let $F = f(X)$ denote the random event that the model predicts a particular value, and let $\eta(F) = \mathbb{P}(Y = 1 \mid F)$, the success probability conditioned on the prediction. The Brier score can be decomposed[1] as

$$\begin{aligned} \mathrm{BS}(f) &= \mathbb{E}\left[(F - Y)^2\right] \\ &= \mathbb{E}\left[(F - \eta(F))^2\right] + \mathbb{E}\left[\eta(F)(1 - \eta(F))\right]. \end{aligned} \quad (1)$$

The first term relates to *calibration*: the discrepancy between the predicted success probabilities and the true conditional event frequencies. A classifier is perfectly calibrated if $F = \eta(F)$. In practice, calibration is estimated by binning predictions into intervals $\{B_k\}_{k=1}^K$ and comparing predicted confidence with empirical frequencies. Let $\bar{f}_k$ be the mean prediction in bin $k$, $\bar{y}_k$ the empirical positive rate, and $n_k$ the bin size. Then the empirical calibration error takes the form $\frac{1}{N}\sum_{k=1}^K n_k(\bar{f}_k - \bar{y}_k)^2$, which is a binned estimator of $\mathbb{E}[(F - \eta(F))^2]$. The commonly used Expected Calibration Error (ECE) replaces the squared deviation by absolute deviation, $\mathrm{ECE}(f) = \sum_{k=1}^K \frac{n_k}{N}|\bar{f}_k - \bar{y}_k|$. It is important to note that calibration does not relate to the accuracy of the predictor. For example, a model that outputs $f = \mathbb{E}[Y]$ independently of $X$ is perfectly calibrated.

The second term in the Brier score decomposition (1) relates to accuracy: it measures how informative the score is about the label. This term is small when predictions induce groups with nearly deterministic outcomes ($\eta(F) \approx 0$ or 1), and large when outcomes remain ambiguous ($\eta(F) \approx 1/2$). In practice, accuracy is typically measured by the ROC AUC.

*Calibration* of a classifier refers to the process of taking a trained classifier $f$ and an i.i.d. calibration dataset $\mathcal{D}_{\mathrm{cal}} = \{(x_i, y_i)\}_{i=1}^{N_c} \sim \mathcal{P}^{N_c}$ and computing a classifier $f'$ that is better calibrated. For neural network classifiers, $f(x) = \sigma(logit(x))$, where $logit(x)$ is the output of the

neural network and $\sigma$ is a sigmoid function. An effective calibration method is temperature scaling (Guo et al., 2017), yielding $f'(x) = \sigma(a \cdot logit(x) + b)$, where $a, b$ are calculated by maximizing likelihood on $\mathcal{D}_{\mathrm{cal}}$.

## 2.2. POMDPs

A Partially Observable Markov Decision Process (POMDP, Åström, 1965) is a popular framework for sequential decision making. It is defined by a tuple $(\mathcal{S}, \mathcal{X}, \mathcal{A}, \mathcal{P}, \mathcal{O}, \mathcal{R}, T)$ where $\mathcal{S}$ is a space of states, $\mathcal{X}$ is a space of observations, $\mathcal{A}$ is a space of actions, $\mathcal{P} : \mathcal{S} \times \mathcal{A} \to \Delta^{\mathcal{S}}$ is a transition model, $\mathcal{O} : \mathcal{S} \to \Delta^{\mathcal{X}}$ is an observation model, $\mathcal{R} : \mathcal{S} \times \mathcal{A} \to \mathbb{R}_+$ is a reward model, and $T \in \mathbb{N}$ is a time horizon.

At each step $t < T$, the POMDP is in a state $s_t \in \mathcal{S}$. The agent observes $x_t \sim \mathcal{O}(s_t)$ and takes an action $a_t \in \mathcal{A}$. The process transitions to $s_{t+1} \sim \mathcal{P}(s_t, a_t)$ and the agent collects a reward $r_t \sim \mathcal{R}(s_t, a_t)$. The process ends in state $s_T$. We denote $h_T := (x_1, a_1, r_1, \ldots, x_T) \in \mathcal{H}$ the *trajectory* of the process and by $h_t$ its prefix of $t < T$ steps.[2]

The agent selects actions according to a policy $\pi : \mathcal{H} \to \Delta^{\mathcal{A}}$ with the goal of maximizing the expected cumulative future rewards. The latter is defined through a *value function*

$$V_i^\pi(h_i) = \mathbb{E}_{h_T \in \mathcal{H}}^\pi \left[ \sum_{t=i}^{T-1} r_t + r_T \,\middle|\, h_i \right],$$

where the expectation is over the action sampled from $\pi$, and states, rewards, and observations sampled from the POMDP. Thus, the agent's objective can be written as $\max_\pi V_t^\pi(h_t), \forall h_t \in \mathcal{H}$. It is typically assumed that the cumulative reward is bounded $V_t^\pi(h_t) \in [0, R_{\max}]$. Finally, we can define an *action value function* (a.k.a. Q function) as

$$Q_i^\pi(h_i, a_i) = \mathbb{E}_{h_T \in \mathcal{H}}^\pi \left[ \sum_{t=i}^{T-1} r_t + r_T \,\middle|\, h_i, a_i \right].$$

## 3. Related Work

**Calibration in Sequential Settings.** In online forecasting, calibration is defined over sequential rounds and analyzed under minimal or adversarial assumptions on the data-generating process (Qiao & Zheng, 2024). Our setting considers predictions along a single episodic trajectory with delayed terminal feedback. In *temporal calibration* (Leathart & Polaczuk, 2020), predictions at different time steps are treated independently, applying post-hoc calibration procedures across temporal slices. In survival analysis (Haider et al., 2020), time-dependent Brier scores are used to evaluate the *time until a failure event*, different from our task of success evaluation. In large language models (LLMs), uncertainty estimation and calibration are inherently sequential, and important for detecting hallucinations (Huang et al.,

---

[1]This decomposition is based on Murphy (1973). A self-contained proof is given in Apx. C.

[2]We denote as $\mathcal{H} := \{\emptyset, \mathcal{H}_1, \ldots \mathcal{H}_T\}$ the power set of the trajectories up to length $T$, where $\mathcal{H}_t := (\mathcal{X} \times \mathcal{A} \times \mathbb{R}_+)^t$.

2024; Shorinwa et al., 2025; Ren et al., 2023; Xiong et al., 2023; Bar-Shalom et al., 2025). In contrast, our work concerns decision making in *dynamic environments*.

In reinforcement learning (RL, Mannor et al. 2026), predicting terminal success corresponds to value estimation, and temporal difference (TD) methods are standard. To the best of our knowledge, the connection with sequential calibration, through the Brier score, is novel. Concurrently with our work, van der Laan & Kallus (2025) connected value learning to calibration, in the context of offline RL. Our work differs in that we consider calibrating a pretrained policy, show a connection between the Brier score and value learning, and demonstrate empirical results on VLAs.

**Calibration of VLAs.** VLAs provide generalist robot policies. Recent architectures including RT-2 (Brohan et al., 2022), Octo (Octo Model Team et al., 2024), OpenVLA (Kim et al., 2024), $\pi_0$ (Black et al., 2024), and $\pi_0$-FAST (Pertsch et al., 2025), map visual observations and textual instructions to low level robot actions, promising zero shot generalization to unseen tasks. However, current VLAs exhibit limited success on tasks *unseen* during training/fine tuning, with reported success rates ranging between 30% and 60% (Kim et al., 2024; O'Neill et al., 2024; Black et al., 2024), motivating study of failure detection.

Zollo & Zemel (2025) first studied confidence calibration in VLAs. They proposed recalibrating action-level confidence estimates using post-hoc methods such as Platt scaling and temperature scaling, using episode-level outcome labels. However, their recalibration operates on individual action confidence scores without modeling calibration dynamics across the trajectory. In contrast, we argue that calibration should be considered across time, investigate more expressive recalibration architectures that aggregate information over the rollout, and additionally address VLAs that do not expose action probabilities.

The most closely related state-of-the-art method is SAFE (Gu et al., 2025), which trains a single cross-task failure detector on internal VLA features and applies conformal prediction (Diquigiovanni et al., 2021) for thresholded decisions. While SAFE does not explicitly target calibration, the method has strong similarities with ours and can be interpreted as an implicit calibration procedure. Nonetheless, we improve upon SAFE by rigorously formulating the sequential calibration problem, proposing a TD calibration method, and showing that our method outperforms SAFE using exactly their features. In addition, while Gu et al. (2025) concluded that VLA action probabilities are insufficient for success prediction, we show that when calibrated using our method, action probabilities yield competitive performance.

Other existing approaches, which SAFE outperforms, primarily consist of unsupervised out-of-distribution (OOD)

detection methods (Xu et al., 2025; Sinha et al., 2024; Wong et al., 2022; Majumdar et al., 2025), which can be overly restrictive for generalist policies, and supervised, task-specific failure detectors (Gokmen et al., 2023; Xie et al., 2022; Farid et al., 2022; Ablett et al., 2020).

## 4. Problem Formulation

We are given a POMDP policy $\pi$. Our goal is to understand, during the run of the policy, whether it will succeed or fail in its task. While this problem has been studied before in various forms (see Section 3), in the following we formally define it in a POMDP setting. As we shall later see, this formulation will allow us to propose new algorithms.

We begin by defining a successful outcome. We say that a trajectory $h_T$ is successful when the cumulative reward $R(h_T) = \sum_{t=1}^{T} r_t$ surpasses a threshold $c \in [0, R_{\max}]$:

$$Y(h_T) = \mathbb{I}\{R(h_T) \geq c\} \in \{0, 1\}. \tag{2}$$

Thus, success depend on the entire trajectory of decisions, which are dependent. Note that for sparse binary rewards $Y(h_T) = R(h_T)$. We would like to evaluate success *during* the run of the policy, and we henceforth define a sequential counterpart to the conventional Brier score (Brier, 1950) defined in Section 2.1. Let $f : \mathcal{H}_t \to [0, 1]$ denote a success prediction function given the history $h_t$.

**Definition 3.** The *sequential Brier score* of a function $f$ is

$$\mathrm{BS}_{\mathrm{seq}}(f, t) := \mathbb{E}^{\pi}_{h_t \in \mathcal{H}_t}\left[\mathbb{E}^{\pi}_{h_T \in \mathcal{H}_T}\left[(f(h_t) - Y(h_T))^2 \,\middle|\, h_t\right]\right] \tag{3}$$

where the expectations are taken over trajectories $h_T$ and their prefix $h_t$ induced by $\pi$.

The sequential Brier score can be empirically estimated from a dataset of trajectories and success pairs $\mathcal{D}_{\mathrm{cal}} = \{h_T^i, y^i\}_{i=1}^{N}$, where $y^i = Y(h_T^i)$ and trajectories $h_T^i$ are sampled with $\pi$, as follows.

$$\widehat{\mathrm{BS}}_{\mathrm{seq}}(f, t) = \frac{1}{N} \sum_{i=1}^{N} \left(f(h_t^i) - y^i\right)^2. \tag{4}$$

Similarly to the conventional Brier score, the sequential Brier score relates to the calibration and accuracy of the success predictor by its two-component decomposition. Let $F_t = f(h_t)$ denote the random event that the model predicts a particular value at time $t$, and let $\eta(F_t) = \mathbb{P}(Y(h_T) = 1 \mid F_t)$, the success probability conditioned on the prediction at time $t$. The following decomposition holds:

$$\mathrm{BS}_{\mathrm{seq}}(f, t) = \mathbb{E}^{\pi}\left[(F_t - Y(h_T))^2\right] \tag{5}$$
$$= \mathbb{E}^{\pi}\left[(F_t - \eta(F_t))^2\right] + \mathbb{E}^{\pi}\left[\eta(F_t)(1 - \eta(F_t))\right],$$

where the first term is referred to as *sequential calibration*, and the second is *accuracy*. Thus, minimizing the sequential

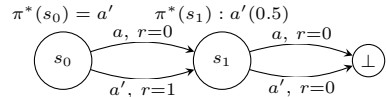

*Figure 2.* Two-step MDP from Example 4.1.

Brier score leads to desired success predictors that are both calibrated and accurate.

Our focus in this work is on finding a success predictor that yields a low Brier score, by computing $f$ from the calibration dataset $\mathcal{D}_{cal}$. Relating to calibration of classifiers (Section 2), we are particularly interested in whether the policy's action probabilities $\pi(h_0), \ldots, \pi(h_t)$ could be calibrated to yield satisfactory performance in success prediction. We therefore introduce the following definition.

**Definition 4.** A success predictor $f$ that depends only on action probabilities, i.e., $f(h_t) = f(\pi(h_0), \ldots, \pi(h_t))$, is termed a *black box success predictor*.

Black box success predictors can be employed when only action probabilities are available - a common setting for commercial foundation models. In contrast, we refer to predictors that require internal access to the VLA model, such as feature activations, as *white box* success predictors.

In the following, we motivate calibration using the sequential Brier score, and explore edge cases. We begin with understanding the importance of the task success labels $y^i$ in the calibration dataset. One may hope that a set of trajectories $h_T^* = (x_0, a_0^*, \ldots, x_{T-1}, a_{T-1}^*)$, which contain expert actions to states visited by the policy $\pi$, are sufficient for sequential calibration. The following example shows that we should not expect this to work well in general.

**Example 4.1.** Consider an MDP described in Fig.2 with horizon $T = 2$ and two actions, $a$ and $a'$. The initial state is $s_0$, and $\mathcal{R}(s_0, a) = 0$, $\mathcal{R}(s_0, a') = 1$. At time step $t = 1$ the state transitions deterministically to $s_1$, and $\mathcal{R}(s_1, a) = \mathcal{R}(s_1, a') = 0$. The terminal reward is 0. Training data is generated by the following optimal policy $\pi^*(s_0) = a'$ w.p. 1, $\pi^*(s_1) = a'$ w.p. 0.5, and assume that the learned policy is $\pi = \pi^*$. For each time step, $\pi$ minimizes the conventional Brier score for individual action prediction. However, for a threshold $c \leq 1$ we have that $Y(h_T) = 1$ for all trajectories, while $\pi(s_1) = 0.5$.

As Example 4.1 shows, in a sequential task some actions may be *uncorrelated* with task success, making action probabilities alone-even if coming from an optimal policy-irrelevant as success predictors. To connect this example to a practical VLA setting, when the robot is moving without grasping anything, the gripper action is irrelevant for task success; if a policy is trained to imitate experts that randomly open and close the gripper, it may exhibit high uncertainty for irrelevant actions. Calibrating to task success data is therefore important.

We next discuss the worst sequential Brier score that we can expect in practice. Given some partial trajectory $h_t$, we have that $\min_f \mathbb{E}^\pi\left[\left(f - Y(h_T)\right)^2 \mid h_t\right] = \mathbb{E}^\pi[Y(h_T)\mid h_t]$.

**Remark 4.2.** Let $\mathbb{E}^\pi[Y(h_T)]$ denote the average task success rate. Any information that exists in the trajectory may be used to obtain a lower Brier score.

## 5. Method

In this section, we devise a method for sequential calibration. The previous discussion clarifies the relation between sequential calibration and prediction of the task success, through the minimization of the sequential Brier score. The way task success is defined in our setting, i.e., whether the total reward collected during a trajectory surpasses a given threshold (see Eq. 2), essentially implies that we can predict task success by predicting future rewards. The problem of future reward prediction is a staple in the RL literature (Mannor et al., 2026). The following theorem formalizes the connection between the two problems.

**Theorem 5.1.** *The future rewards predictor $\hat{f}(h_t)$ that minimizes the Sequential Brier Score*

$$\mathbb{E}^\pi_{h_t \in \mathcal{H}_t}\left[\mathbb{E}^\pi_{h_T \in \mathcal{H}_T}\left[(\hat{f}(h_t) - R(h_T))^2 \mid h_t\right]\right]$$

*is given by:*

$$\hat{f}(h_t) = \mathbb{E}^\pi_{h_T \in \mathcal{H}_T}[R(h_T) \mid h_t] = \sum_{i=1}^{t-1} r(h_i) + Q_t^\pi(h_t, a_t).$$

Theorem 5.1 fundamentally links our original problem of calibrating a given policy $\pi$ with estimating its value function $Q^\pi$. In other words, *learning the value function is equivalent to learning a calibrated predictor of eventual success*. Value estimation is a well-studied problem in RL (Mannor et al., 2026) and allows us to add the family of value estimation algorithms to the calibration toolkit.

In this paper, we consider access to a policy $\pi$ and a set of $N$ labeled trajectories $\mathcal{D}_{cal} = \{(h_T^i, y^i)_{i=1}^N\}$ generated with the policy, where the label denotes whether the trajectory was successful. The value estimation algorithm fits a parametric function $f_\theta : \mathcal{H}_t \to [0, 1]$ to the data by minimizing a loss

$$\mathcal{L}(\theta) := \frac{1}{NT}\sum_{i=1}^N \sum_{t=1}^T \ell(f_\theta(h_t^i), y^i).$$

In RL, the single-step loss $\ell(f_\theta(h_t^i), y^i)$ typically takes two forms. One is called *Monte Carlo* (MC), and prescribes to solve a classification or regression problem between $h_t^i$ and $y^i$. The other is *Temporal Difference* (TD), which bootstraps the value between two consecutive steps as

$$\ell(f_\theta(h_t^i), y^i) = \begin{cases} \left(f_\theta(h_t^i) - f_\theta(h_{t+1}^i)\right)^2 & t < T \\ \left(f_\theta(h_t^i) - y^i\right)^2 & t = T \end{cases}. \quad (6)$$

**Algorithm 1** TDQC

---

**input** Policy $\pi$, calibration dataset $\mathcal{D}_{\text{cal}} = \{(h_T^i, y^i)_{i=1}^N\}$

1: Initialize network weights $f_\theta$, $f_{\theta^-}$
2: **for** until convergence **do**
3:     Sample $h_T^i$ uniformly from $\mathcal{D}_{\text{cal}}$
4:     Compute the TDQC loss using a target network

$$G(\theta) = \sum_{t=1}^{T-1} (f_\theta(h_t^i) - f_{\theta^-}(h_{t+1}^i))^2 + (f_\theta(h_T^i) - y^i)^2$$

5:     Update $\theta \leftarrow \theta - \alpha \nabla G(\theta)$
6:     Periodically update target network $\theta^- \leftarrow \theta$
7: **end for**

---

For training stability of TD loss, we used updates with a target network Mnih et al. (2015). The target network parameters $\theta^-$ are updated with the current parameters $\theta$ every $C$ steps and are held fixed between individual updates. While previous work (Gu et al., 2025) employs an MC loss, TD has not been considered in the context of calibration. TD is the common practice in RL (e.g., Haarnoja et al., 2018) for its well-established practical and theoretical benefits (Kearns & Singh, 2000). In the experiments, we will show that similar benefits extend to the calibration setting. Before that, in Algorithm 1, we provide a brief summary of our method, which we call **T**emporal-**D**ifference **Q**-based **C**alibration, TDQC for short. Then, we comment on how to extract information from the history in practice, e.g., when running VLA policies, and how having a calibrated policy opens the door to failure detection and other applications.

**Extracting the History.** In principle, the history $h_t$ should contain all signals available at time $t$ that are informative about future success, e.g., the current observation, past actions, and any internal state carried by the policy. In practice, we consider two lightweight ways to extract $h_t$ from a VLA.

For *token-based* VLA architectures, actions are discretized into tokens, and the policy outputs a categorical distribution over tokens at each step. This design is used by RT-2 (Zitkovich et al., 2023), OpenVLA (Kim et al., 2024), NORA (Hung et al., 2025) and Uni-VLA (Wang et al., 2025). Since these models resemble LLM-style decoders, we can adopt the common heuristic of using token probabilities and use a *black-box success predictor* (Def. 4).

More generally, we can define $h_t$ via the policy's internal representations, such as hidden states from intermediate layers. Using internal features as inputs to an auxiliary predictor is standard in the LLM calibration and *probing* literature (Azaria & Mitchell, 2023; Zou et al., 2023; Kossen et al., 2024), but requires access to the model's weights.

### 5.1. Early Stopping with Conformal Prediction

When a policy $\pi$ is deployed in safety-critical scenarios, we may want to exploit $f_\theta$ to stop early trajectories that have low probability of success, avoiding unnecessary damage to the system and its surroundings in failed attempts.

To this purpose, we use $f_\theta$ as a confidence score for future success, which is equivalent to our original definition of predicting future rewards for binary tasks. Then, $f_\theta$ can be used to trigger early stopping when the predicted probability of success is low. This can be made formal using *conformal prediction* (CP, Shafer & Vovk, 2008; Jazbec et al., 2024; Ringel et al., 2024).

Following Gu et al. (2025), we treat $1 - f_\theta$ as a failure predictor and raise a failure flag whenever $1 - f_\theta > \delta_t$, where $\delta_t$ is a time-varying upper threshold. To calculate that threshold at each time step we first set a significance level $\alpha \in (0, 1)$, which determines how conservative we want to be with the early stopping, and we separate a set of successful trajectories from the data before training, to avoid biasing the procedure. On this validation set, we invoke a *functional* conformal prediction (CP) routine (see Algorithm 2 in Apx. D and Diquigiovanni et al. 2021 for details) to compute a time-varying confidence upper bound $\{\delta_t\}_{t=1}^T$. Finally, while the policy is deployed, we process the current trajectory $h_t$ with $f_\theta$ and compare the resulting score with $\delta_t$. Whenever $1 - f_\theta(h_t) > \delta_t$ we stop $\pi$. The described procedure is reported in Algorithm 3. Under the exchangeability assumption (Vovk et al., 2005), the CP band guarantees that, for a new *successful* rollout, $1 - f_\theta$ remain below the threshold for all time $t$ w.p $1 - \alpha$.

### 5.2. Application to Test-Time Guided Action Search

As an example downstream application of our calibration method, we consider action search at test time. Standard VLA models typically execute the single most probable action predicted by their policy. However, this greedy approach can lead to sub-optimal decisions, especially when attempting to generalize to new environments.

Related LLM-based approaches combine search with process reward models (PRMs), which calculate the quality or correctness of intermediate reasoning steps (Lightman et al., 2023; Zhang et al., 2025). Analogously, to exploit calibration to this end, we propose to use the learned Q-function $f_\theta$ to guide action search toward trajectories that are more likely to succeed, inspired by RL methods that employ search at test time (Silver et al., 2016). Furthermore, when $f_\theta$ indicates a high confidence that the trajectory will succeed, we can save compute resources and skip the search (this is the converse of early stopping, cf. Section 5.1).

The pseudocode of the procedure is reported in Algorithm 4. Note that we assume access to an accurate forward model of the system dynamics $W = P(x_{t+1}|h_t, a_t)$, which is easily available in simulated domains. For real-world domains, a world model of the dynamics may be learned (Schrittwieser

et al., 2020); we leave such investigation for future work.

At each time step, given a VLA policy $\pi$ and a simulator $W$, we sample from the policy $M$ possible action candidates. For each candidate action $i$, we simulate a one-step look-ahead to observe the resulting next observation $x_{t+1}^{(i)} \sim W(x_t, a_t^{(i)})$ , and subsequently select the next action using the standard model's greedy selection strategy $\max_{a'} \pi(x_{t+1}, a')$. We then evaluate each resulting observation using $Q = f_\theta(x_{t+1}, a_{t+1})$ and choose the action with highest score $a^* = \max_{a'} Q(x_{t+1}, a')$. We emphasize that $f_\theta$ parameters are frozen and used only for test-time guided search. To reduce test-time compute using the *calibrated* model, we apply the action search only when the predicted Q value exceeds a threshold $\bar{T}$.

# 6. Experiments

In this section, we empirically validate the connection between calibrating a policy $\pi$ and estimating its value $Q_t^\pi$. We test TDQC, our *sequentially calibrated* success predictor in sequential tasks. We aim to answer the following: *(1)* TD loss improves calibration and failure detection results? *(2)* Can black-box based calibration yield competitive results? *(3)* Can a calibrated model be used to improve success rate using test-time guided action search?

## 6.1. Vision-Language-Action Models

We evaluate four state-of-the-art VLA policies: OpenVLA (Kim et al., 2024), UniVLA (Wang et al., 2025), $\pi_0$ (Black et al., 2024), and $\pi_0$-FAST (Pertsch et al., 2025). These models span different action parameterizations, which directly changes what confidence signals can be extracted from their outputs. OpenVLA discretizes continuous action into tokens and exposes the corresponding token probabilities, which we use as the action probabilities. $\pi_0$-FAST and UniVLA also employs a tokenized representation and apply Discrete Cosine Transform (DCT) to convert continuous action sequences into discrete action tokens. Consequently, token probabilities do not match with the per-dimension continuous action and cannot be directly interpreted as such. The $\pi_0$ VLA represents continuous action with a flow matching model, rather than per-step token probabilities. All models are open-sourced, enabling access to internal features.

Following Def. 4, we refer to methods that predict success using only the model's observable outputs as *black-box*. For OpenVLA, our black-box approach uses the discrete action probabilities directly. For $\pi_0$-FAST and UniVLA, the FAST action tokenizer decodes tokens into discrete actions, each with logits spanning the decoder's vocabulary size. Our black-box approach takes these logits as input; see Apx. F.3 for more details on the extraction of those probabilities. For all white-box predictors, we use as input the features suggested in SAFE (Gu et al., 2025).

## 6.2. Benchmarks

**LIBERO** (Liu et al., 2023) is a popular simulated benchmark for VLA evaluation. We evaluated OpenVLA, $\pi_0$-FAST and $\pi_0$ on LIBERO-10 suite, which contains 10 long horizon tasks with diverse objects, layouts, and instructions, and is considered the *most challenging* suite (Gu et al., 2025; Zollo & Zemel, 2025). Episodes terminate upon task completion or after a timeout (failure). We let $T_i$ denote rollout $i$'s stopping time: either the time of task completion, or the timeout duration on failure (see Gu et al. (2025) and Apx. F.4 for more details).

**Real-World WidowX** We consider the WidowX data and OpenVLA checkpoints published in Gu et al. (2025). The dataset contains 532 rollouts on 8 pick-and-place tasks. *All tasks* have rollouts with exactly $T_i = 50$ steps; if the policy succeeds earlier, the robot keeps acting in the environment until reaching $T_i$ (see Gu et al. 2025 and Apx. F.4).

**Real-World Franka** We consider the Franka Emika Panda Robot in Gu et al. (2025) using $\pi_0$-FAST-DROID checkpoints (Pertsch et al., 2025). The dataset contains 13 pick and place tasks, each with 30 successful and 30 failed rollouts. Each *task* $k$ has the same number of steps $T_k$, regardless if they failed earlier or not, thus $T_i = T_k$ for each rollout $i$ in task $k$ (see Gu et al. 2025 and Apx. F.4).

## 6.3. Baselines

**Static Predictors.** We compared Max Prob, Avg Prob and Running Avg Prob presented in Zollo & Zemel (2025), Avg Entropy and Running Avg Entropy from Gu et al. (2025). These methods use the raw action probabilities per time step or apply simple, hand-crafted transformations, such as maximum or average over both action dimension and time, on the raw probabilities and on the probabilities' entropy. More details and formulation appear in the Apx. F.5.

**Trained Predictors.** We used the baselines from Gu et al. (2025), which provide, to the best of our knowledge, the state-of-the-art failure detection baselines for VLA models. SAFE-MLP uses a small MLP network and takes as input the internal feature vectors (hidden state) $e$ from the model, i.e., $h_t = e_t$. SAFE-RNN applies a linear projection layer to the input and then an LSTM (Hochreiter & Schmidhuber, 1997). For the evaluation, we followed the same networks and parameters used in Gu et al. (2025) as the $f$ in SAFE methods. Note that both SAFE baselines use hidden states as input, hence are white-box methods.

Apx. F.6 and Gu et al. (2025) detail how the hidden features are extracted and on the architectures of both models. Here we note that the SAFE-MLP loss aggregates the scores over time $\mathcal{L}_{\text{MLP}} = \sum_i [y_i \sum_t (t - f_\theta(e_t)) + (1 - y_i) \sum_t f_\theta(e_t)]$ which induces a monotone, time-indexed scoring scheme rather than

| | VLA Model | OpenVLA | | OpenVLA | | UniVLA | | $\pi_0$-FAST | | $\pi_0$-FAST | | $\pi_0$ | |
| | Benchmark | LIBERO | | WidowX | | LIBERO | | LIBERO | | Franka | | LIBERO | |
| | Eval Task Split | Seen ↓ | Unseen ↓ | Seen ↓ | Unseen ↓ | Seen ↓ | Unseen ↓ | Seen ↓ | Unseen ↓ | Seen ↓ | Unseen ↓ | Seen ↓ | Unseen↓ |
|---|---|---|---|---|---|---|---|---|---|---|---|---|---|
| | Max prob. | 0.395 | 0.390 | 0.572 | 0.579 | 0.909 | 0.899 | 0.320 | 0.318 | 0.331 | 0.323 | – | – |
| | Avg prob. | 0.348 | 0.364 | 0.275 | 0.282 | 0.529 | 0.532 | 0.212 | 0.218 | 0.290 | 0.294 | – | – |
| Static Predictors | Running Avg prob. | 0.338 | 0.356 | 0.255 | 0.257 | 0.543 | 0.544 | 0.244 | 0.238 | 0.359 | 0.361 | – | – |
| | Avg entropy | 0.306 | 0.313 | 0.414 | 0.426 | 0.406 | 0.391 | 0.209 | 0.222 | 0.281 | 0.281 | – | – |
| | Running Avg entropy | 0.265 | 0.273 | 0.435 | 0.432 | 0.343 | 0.330 | 0.279 | 0.264 | 0.341 | 0.339 | – | – |
| | *SAFE-RNN* | 0.204 | 0.255 | 0.169 | 0.213 | 0.124 | 0.162 | 0.106 | 0.148 | 0.220 | 0.288 | 0.123 | 0.172 |
| Learned Predictors (white box) | *SAFE-RNN-TDQC (Ours)* | 0.197 | 0.218 | **0.096** | **0.153** | **0.064** | **0.100** | **0.103** | 0.163 | **0.150** | **0.215** | **0.061** | **0.097** |
| | *SAFE-MLP BCE* | 0.192 | 0.231 | 0.127 | 0.164 | 0.091 | 0.158 | **0.103** | 0.162 | 0.206 | 0.248 | 0.075 | 0.137 |
| | *SAFE-MLP-TDQC (Ours)* | 0.195 | 0.229 | 0.130 | 0.169 | 0.066 | 0.131 | 0.109 | 0.150 | 0.210 | 0.229 | 0.068 | 0.128 |
| Learned predictors (black box) | *RNN-BCE* | 0.199 | 0.206 | 0.301 | 0.344 | 0.138 | 0.152 | 0.122 | 0.158 | 0.237 | 0.243 | – | – |
| | *RNN-TDQC (Ours)* | **0.191** | **0.197** | 0.156 | 0.192 | 0.100 | 0.107 | 0.105 | **0.141** | 0.204 | 0.228 | – | – |

*Table 1.* Brier Score results on simulation and real robot experiment. Results are averaged over 21 seeds which determine different train-test splits of the tasks. "−" indicates that the Brier score can't be calculated on the method. The **best** performing methods are highlighted in the table. Our method, TDQC achieves lowest sequential Brier scores in all benchmarks.

| | VLA Model | OpenVLA | | OpenVLA | | UniVLA | | $\pi_0$-FAST | | $\pi_0$-FAST | | $\pi_0$ | |
| | Benchmark | LIBERO | | WidowX | | LIBERO | | LIBERO | | Franka | | LIBERO | |
| | Eval Task Split | Seen ↑ | Unseen ↑ | Seen ↑ | Unseen ↑ | Seen ↑ | Unseen ↑ | Seen ↑ | Unseen ↑ | Seen ↑ | Unseen ↑ | Seen ↑ | Unseen↑ |
|---|---|---|---|---|---|---|---|---|---|---|---|---|---|
| | Max prob. | 54.64 | 55.78 | 53.25 | 53.20 | 50.00 | 50.00 | 61.75 | 63.49 | 48.61 | 46.64 | – | – |
| | Avg prob. | 47.08 | 48.09 | 47.47 | 48.30 | 42.96 | 40.25 | 47.36 | 48.09 | 49.45 | 48.03 | – | – |
| Static Predictors | Running Avg prob. | 49.19 | 47.72 | 49.20 | 46.37 | 43.05 | 40.29 | 53.95 | 55.68 | 52.95 | 49.98 | – | – |
| | Avg entropy | 46.81 | 46.75 | 50.19 | 49.36 | 41.34 | 47.36 | 45.42 | 46.30 | 49.28 | 49.07 | – | – |
| | Running Avg entropy | 50.48 | 48.09 | 46.16 | 43.99 | 51.63 | 56.71 | 53.78 | 55.44 | 51.12 | 48.78 | – | – |
| | *SAFE-RNN* | 72.30 | 69.04 | 75.95 | 70.00 | 74.48 | 68.13 | 91.26 | **85.88** | 70.85 | 56.69 | 79.96 | 65.03 |
| | *SAFE-RNN-TDQC (Ours)* | 71.67 | 65.12 | 84.01 | 70.76 | 73.83 | 64.25 | **92.03** | 85.49 | **79.89** | **68.43** | 88.66 | **82.94** |
| Learned Predictors (white-box) | *SAFE-MLP* | 73.56 | 69.53 | **88.23** | **83.18** | 74.39 | 63.80 | 79.00 | 68.36 | 79.15 | 63.94 | 86.33 | 79.75 |
| | *SAFE-MLP-BCE* | 72.66 | 64.99 | 85.38 | 71.43 | **78.83** | 69.74 | 91.82 | 85.50 | 73.82 | 58.83 | **89.71** | 80.53 |
| | *SAFE-MLP-TDQC (Ours)* | 71.22 | 60.09 | 82.33 | 70.64 | 76.55 | 66.71 | 90.25 | 84.44 | 61.95 | 51.22 | 86.57 | 72.07 |
| Learned predictors (black box) | *RNN-BCE* | 72.70 | 72.28 | 69.69 | 67.09 | 63.44 | 54.37 | 87.67 | 82.53 | 59.32 | 53.26 | – | – |
| | *RNN-TDQC (Ours)* | **74.20** | **72.72** | 78.90 | 72.97 | 72.30 | **70.17** | 87.51 | 83.39 | 64.19 | 57.37 | – | – |

*Table 2.* ROC-AUC results on simulation and real robot experiment. Results are averaged over 21 seeds which determine different train-test splits of the tasks. "−" indicates that the ROC-AUC can't be calculated on the method. The **best** performing methods are highlighted in the table. Our method, TDQC achieves SOTA results on unseen tasks in 4 benchmarks out of 6.

a calibrated probability. Its uncertainty estimation at each time step is between $f(h_{t-1}) \leq f(h_t) \leq t$ and cannot be treated as a probability. Post-hoc normalization would be arbitrary and distort comparisons. Therefore, we keep the same MLP architecture but replaced the loss with a standard binary cross-entropy objective, yielding SAFE-MLP BCE.

We compared our method TDQC in two settings: (1) We use black-box success predictor (Def. 4) for OpenVLA, UniVLA and $\pi_0$-FAST as described in Section 6.1 combined with TDQC loss (Alg. 1). (2) We incorporate our TDQC loss (Alg. 1) on SAFE baselines where $h_t$ is the policy's internal features at time step $t$. Note that both settings are alternative ways for approximating $Q_t^\pi$.

### 6.4. TD loss improves calibration and failure detection

**Calibration.** To test the VLA policies' calibration, we measure the *sequential Brier score* (Definition 3), which measures calibration and accuracy as shown in Eq. 5. Each rollout $i$ has a horizon $T_i$, and produces an episode-level binary success label $Y(h_T^i)$. To compare calibration across rollouts with different lengths, we report sequential Brier score in different *quantiles* $q \in [0, 1]$, where $q = t/T_i$ denotes a normalized time within rollout $i$.

Table 1 shows that across models and benchmarks, *TDQC improves calibration relative to non-TDQC variants*, where SAFE-RNN-TDQC performs the best or on par for all $\pi_0$ model variants. Moreover, Fig 1 reports sequential Brier scores at different time quantiles within an episode. Early

in the rollout, when limited task information is available, TD-based predictors perform comparably to binary cross-entropy baselines. As time advances, sequential Brier scores decrease for all methods, reflecting that explicitly modeling the temporal structure yields more accurate success probability estimates than step-wise or purely supervised training.

The dotted horizontal line in Figure 1 represents the Brier score of a constant predictor that, instead of using the model's task-specific success estimates, consistently outputs the empirical mean success rate computed over the seen tasks, as described in Section 4.2. Consequently, it constitutes an uninformative baseline. We observe that several BCE-based approaches yield little to no improvement relative to this baseline, whereas most TDQC-based approaches attain progressively lower Brier scores, indicating superior calibration and more informative predictive distributions.

**Failure Detection.** We evaluate failure detection using ROC-AUC, which measures how well a score ranks failed rollouts above successful ones and is widely used for uncertainty quantification in LLMs (Huang et al., 2023; Farquhar et al., 2024; Xia et al., 2025). A failure is flagged once the CP threshold is exceeded. For each task, we compute the minimum rollout length and evaluate ROC-AUC using the maximum prediction up to this timestep.

Table 2 shows that predictors with TDQC loss perform better or on par with the best baselines for both white-box and black-box predictors. Based on Table 1 and Table 2, we conclude that *TDQC sets the state-of-the-art on unseen*

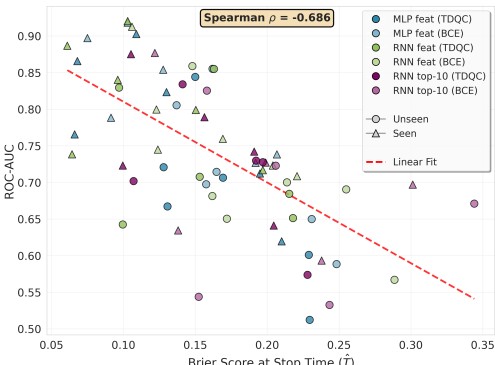

*Figure 3.* **ROC-AUC vs Brier score** over all learned baselines in all benchmarks at the minimum rollout length. Points are grouped by method and split, with a dashed linear fit; the Spearman correlation is $\rho = -0.686$ which indicates high negative correlation.

*tasks on OpenVLA LIBERO, $\pi_0$ LIBERO, UniVLA LIBERO and real robot $\pi_0$-FAST Franka.*

Finally, since ROC-AUC and sequential Brier are related, we observe in Fig. 3 a consistent empirical trend. Methods with lower sequential Brier score tend to achieve higher ROC-AUC, suggesting that better calibrated success probabilities are useful for failure detection. The Spearman correlation of $-0.686$ highlights the strong connection between the two measures. Apx. G.10 examines per-VLA results: all display high correlation, with some VLAs showing stronger correlation, indicating more informative internal representations.

### 6.5. Black-box calibration yield competitive results

To evaluate calibration of action or token probabilities we measured the Brier score for learned black-box methods and static methods. Table 1 shows that black-box predictors achieve best or competitive results on all LIBERO benchmarks, when trained using TDQC. Further, TDQC significantly improves performance of black-box predictors across all benchmarks. In comparison, static baselines, which also use action probabilities as input but are not calibrated by learning from trajectory success labels, display significantly worse performance. These results suggest that the right calibration technique (here, TDQC) can extract useful signal even in a black-box setting.

### 6.6. Application to guided test-time action search

To evaluate the guided test-time search application described in Section 5.2, We measured OpenVLA's success rates on three held-out LIBERO-10 tasks, with 50 rollouts per task. We compared the unmodified OpenVLA policy (Baseline) against several value guided search configurations, using Alg. 4. In Fig. 4 we show the success rate relative to the increase in test time compute. *RNN with TDQC or BCE* action selection methods uses the output of the $f_\theta$ network for the guidance *at all time steps* (that is, the threshold in

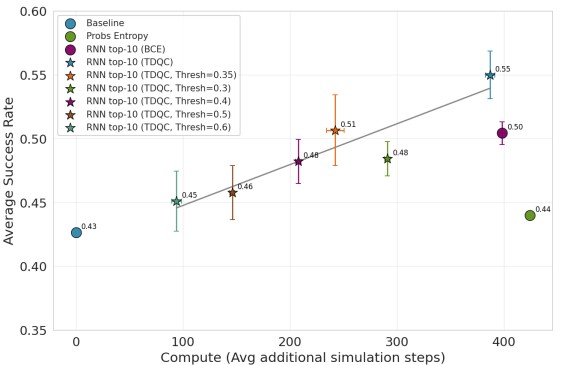

*Figure 4.* **Averaged success rates over test time compute for action selection configurations**. All experiments evaluate 3 unseen tasks from LIBERO-10 taken with OpenVLA, averaged over 3 seeds. RNN-TDQC achieves highest success rates, while RNN-TDQC with thresholds tradeoff success rate and compute.

Algorithm 4 is $\bar{T} = -\infty$). *TDQC - Thresh* method saves compute by applying a confidence threshold, as described in Alg. 4. In this experiment we used a threshold $\bar{T} = 0.35$, which achieved a good tradeoff between success rate and compute. To increase variance in sampled actions, we generated 10 samples per timestep in all value guided methods using a sampling temperature of 1.5.

Fig 4 shows that RNN methods with top-10 probabilities gain significant improvements and outperform the baseline VLA policy, validating the use of the learned scoring function ($f_\theta$) for action selection. Notably, the RNN method trained with TDQC achieves the highest overall performance, reaching 55% success rate on average, an improvement of 13% over the regular baseline. While the BCE loss variant also improves upon the baseline, it peaks lower at 50%. Moreover, applying a threshold to the TDQC method (TDQC, Thresh 0.35) results in a slight performance drop compared to the unconstrained TDQC approach, but saves almost half of the additional compute. Interestingly, we observe a near-linear relation between additional compute and success rate (the gray line in Figure 4 is a fit on all RNN top-10 methods). Additional details are in Apx. G.3.

## 7. Conclusion and Future Works

In this paper, we proposed a novel formulation for calibration in sequential tasks. We showed that predictors calibrated using our temporal-difference based approach achieve SOTA performance in several VLA benchmarks. Importantly, our approach can work with only action probabilities, which is essential for calibration of *black-box* proprietary models accessed via APIs.

We see potential in using calibrated success predictors for action search at test time, in particular, allocating search effort only to "difficult" decisions, can be a promising direction for resource management.

## Acknowledgments

We thank Qiao Gu for helpful conversations and for releasing his code and data for SAFE. This research was funded by the European Union (ERC, Bayes-RL, 101041250). Views and opinions expressed are however those of the author(s) only and do not necessarily reflect those of the European Union or the European Research Council Executive Agency (ERCEA). Neither the European Union nor the granting authority can be held responsible for them. Y. R. was supported by the Israel Science Foundation (ISF grant 729/21). He also acknowledges the additional support from the Career Advancement Fellowship at the Technion.

## Impact Statement

This paper presents work whose goal is to advance the field of Machine Learning. There are many potential societal consequences of our work, none of which we feel must be specifically highlighted here.

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

# A. Limitations and future work

**Limitations.** We emphasize three main limitations. First, we observe that the failure predictor generalizes across unseen tasks within an environment but not across environments, embodiments, or action parameterizations. For instance, we report in Apx. G.2 that transferring the OpenVLA LIBERO-10 predictor directly to the WidowX environment yields only $53.69 \pm 5.07$ ROC-AUC and a sequential Brier score of $0.413 \pm 0.03$. We believe that such broad generalization may require a different approach. Second, our experimental results are performed with binary episodic success; Note however that our formulation in Section 4 accounts for general reward functions and TDQC can easily be extended to such. Third, our guided action-selection operates under a one-step look-ahead. While effective in simulation, this requires environment access at inference time and is therefore not directly deployable in real-world settings without a forward model. Recent advances in world models for robotics (Badithela et al., 2025; Gemini Robotics Team et al., 2026) offer a promising path toward addressing this.

**Future Works.** We conclude with several potential future extensions of our work. Recent advances on world models (Gemini Robotics Team et al., 2026; Badithela et al., 2025) could be exploited to improve uncertainty estimation of robot policies in the real world. Our approach could also be extended for LLM calibration (Orgad et al., 2025; Radharapu et al., 2025), interpreting text generation as a sequential task.

# B. Proof for Theorem 5.1

Fix a policy $\pi$ and consider an episodic process with horizon $T$. Let $h_t$ denote the history available up to time $t$, as defined in Section 2.2. We study the problem of predicting the *future rewards* from partial information: at time $t$ we observe $h_t$ and seek a predictor $\hat{f}(h_t)$ of the total future rewards.

$$R := R(h_T) = \sum_{i=1}^{T} r(h_i), \tag{7}$$

where $r(h_i)$ denotes the (possibly stochastic) reward at step $i$ given the history $h_t$.

Theorem 5.1 states that the predictor minimizing the conditional mean squared error

$$\mathbb{E}_{h_t \in \mathcal{H}_t}^{\pi} \left[ \mathbb{E}_{h_T \in \mathcal{H}_T}^{\pi} \left[ (R(h_T) - \hat{f}(h_t))^2 \mid h_t \right] \right]$$

is given by:

$$\hat{f}(h_t) = \mathbb{E}_{h_T \in \mathcal{H}_T}^{\pi}[R(h_T) \mid h_t] = \sum_{i=1}^{t-1} r(h_i) + Q_t^{\pi}(h_t, a_t)$$

Meaning that, in order to minimize the *sequential Brier Score*, one must approximate the Q-function $Q_t^{\pi}(h_t, a_t)$.

*Proof.* We begin by denote $R(h_T) := R$ for ease of notation. We recall that the optimal estimator in terms of MSE takes the form of $\mathbb{E}_{h_T \in \mathcal{H}_T}[R \mid h_t]$.

Suppose we stop at step $t$ and can only access $h_t$, meaning we don't know $h_T$ and thus, don't have access to $\{r(h_i)\}_{i=t+1}^{T}$, so we'll replace $\mathbb{E}_{h_T \in \mathcal{H}_T}^{\pi}[\sum_{j=t+1}^{T} r(h_j) \mid h_t]$ with $\hat{f}(h_t)$.

$$\mathbb{E}_{h_T \in \mathcal{H}_T} \left[ R \middle| h_t \right] = \mathbb{E}_{h_T \in \mathcal{H}_T}^{\pi} \left[ \sum_{i=1}^{t} r(h_i) + \sum_{j=t+1}^{T} r(h_j) \middle| h_t \right] \overset{\text{markov}}{=}$$

$$\mathbb{E}_{h_T \in \mathcal{H}_T}^{\pi} \left[ \sum_{i=1}^{t} r(h_i) \middle| h_t \right] + \mathbb{E}_{h_T \in \mathcal{H}_T}^{\pi} \left[ \sum_{j=t+1}^{T} r(h_j) \middle| h_t \right] = \sum_{i=1}^{t} r(h_i) + \mathbb{E}_{h_T \in \mathcal{H}_T}^{\pi} \left[ \sum_{j=t+1}^{T} r(h_j) \middle| h_t \right] = \sum_{i=1}^{t} r(h_i) + \hat{f}(h_t)$$

Where the last equality holds since we know the rewards and histories until time step $t$.

Note that:

$$\mathbb{E}^{\pi}_{h_T \in \mathcal{H}_T} \left[ \sum_{j=t+1}^{T} r(h_j) \Big| h_t \right] = Q_t^{\pi}(h_t, a_t) - r(h_t)$$

The amended sequential Brier Score for this case is:

$$\mathbb{E}^{\pi}_{h_t \in \mathcal{H}_t} \left[ \mathbb{E}^{\pi}_{h_T \in \mathcal{H}_T} \left[ \left( R - \sum_{i=1}^{t} r(h_i) - \hat{f}(h_t) \right)^2 \Big| h_t \right] \right]$$

Because we have information until time step $t$, then $\hat{f}(h_t)$ is constant and can be treated as a number. Also, $\mathbb{E}^{\pi}_{h_T \in \mathcal{H}_T} \left[ \sum_{i=1}^{t} r(h_i) \right]$ is a constant and can be outside of the inner expectation. Let's look at the inner expectation:

$$\mathbb{E}^{\pi}_{h_T \in \mathcal{H}_T} \left[ \left( R - \sum_{i=1}^{t} r(h_i) - \hat{f}(h_t) \right)^2 \Big| h_t \right] = \mathbb{E}^{\pi}_{h_T \in \mathcal{H}_T} \left[ R^2 \mid h_t \right] - 2 \cdot \mathbb{E}^{\pi}_{h_T \in \mathcal{H}_T} \left[ R \mid h_t \right] \cdot \sum_{i=1}^{t} r(h_i)$$

$$- 2 \cdot \mathbb{E}^{\pi}_{h_T \in \mathcal{H}_T} \left[ R \mid h_t \right] \cdot \hat{f}(h_t) + 2 \cdot \sum_{i=1}^{t} r(h_i) \cdot \hat{f}(h_t) + \left( \sum_{i=1}^{t} r(h_i) \right)^2 + \hat{f}(h_t)^2$$

We can add and subtract $\left( \mathbb{E}^{\pi}_{h_T \in \mathcal{H}_T} \left[ R \mid h_t \right] \right)^2$ from the equation:

$$\mathbb{E}^{\pi}_{h_T \in \mathcal{H}_T} \left[ R^2 \mid h_t \right] - 2 \cdot \mathbb{E}^{\pi}_{h_T \in \mathcal{H}_T} \left[ R \mid h_t \right] \cdot \sum_{i=1}^{t} r(h_i) - 2 \cdot \mathbb{E}^{\pi}_{h_T \in \mathcal{H}_T} \left[ R \mid h_t \right] \cdot \hat{f}(h_t)$$

$$+ 2 \cdot \sum_{i=1}^{t} r(h_i) \cdot \hat{f}(h_t) + \left( \sum_{i=1}^{t} r(h_i) \right)^2 + \hat{f}(h_t)^2 + \left( \mathbb{E}^{\pi}_{h_T \in \mathcal{H}_T} \left[ R \mid h_t \right] \right)^2 - \left( \mathbb{E}^{\pi}_{h_T \in \mathcal{H}_T} \left[ R \mid h_t \right] \right)^2$$

$$= \text{Var}\left( R \mid h_t \right) - 2 \cdot \mathbb{E}^{\pi}_{h_T \in \mathcal{H}_T} \left[ R \mid h_t \right] \cdot \sum_{i=1}^{t} r(h_i) - 2 \cdot \mathbb{E}^{\pi}_{h_T \in \mathcal{H}_T} \left[ R \mid h_t \right] \cdot \hat{f}(h_t)$$

$$+ 2 \cdot \sum_{i=1}^{t} r(h_i) \cdot \hat{f}(h_t) + \left( \sum_{i=1}^{t} r(h_i) \right)^2 + \hat{f}(h_t)^2 + \left( \mathbb{E}^{\pi}_{h_T \in \mathcal{H}_T} \left[ R \mid h_t \right] \right)^2$$

$$= \text{Var}\left( R \mid h_t \right) + \left( \mathbb{E}^{\pi}_{h_T \in \mathcal{H}_T} \left[ R \mid h_t \right] - \sum_{i=1}^{t} r(h_i) - \hat{f}(h_t) \right)^2$$

Lets break down the term $\mathbb{E}^{\pi}_{h_T \in \mathcal{H}_T} \left[ R \mid h_t \right]$:

$$\mathbb{E}^{\pi}_{h_T \in \mathcal{H}_T} \left[ R \mid h_t \right] = \mathbb{E}^{\pi}_{h_T \in \mathcal{H}_T} \left[ \sum_{i=1}^{t} r(h_i) + \sum_{j=t+1}^{T} r(h_j) \mid h_t \right] = \tag{8}$$

$$\sum_{i=1}^{t} r(h_i) + \mathbb{E}^{\pi}_{h_T \in \mathcal{H}_T} \left[ \sum_{i=t+1}^{T} r(h_i) \mid h_t \right] = \sum_{i=1}^{t} r(h_i) + \mathbb{E}^{\pi}_{h_T \in \mathcal{H}_T} \left[ \sum_{i=t+1}^{T} r(h_i) \mid h_t \right]$$

We can use Equation (8) and the inner expectation looks like:

$$\boxed{\mathbb{E}^{\pi}_{h_T \in \mathcal{H}} \left[ (R(h_T) - \hat{f}(h_t))^2 \mid h_t \right] = \text{Var}\left( R \mid h_t \right) + \left( \mathbb{E}^{\pi}_{h_T \in \mathcal{H}_T} \left[ \sum_{i=t+1}^{T} r(h_i) \mid h_t \right] - \hat{f}(h_t) \right)^2}$$

The $\hat{f}(h_t)$ that will minimize this term is (inverse of law of total expectation):

$$\hat{f}(h_t) = \mathbb{E}^{\pi}_{h_T} \left[ \sum_{i=t+1}^{T} r(h_i) \mid h_t \right] = Q_t^{\pi}(h_t, a_t) - r(h_t)$$

□

## C. Brier Score Decomposition

By adding and subtracting $\eta(F)$ and expanding the square,

$$(F - Y)^2 = (F - \eta(F) + \eta(F) - Y)^2$$
$$= (F - \eta(F))^2 + (\eta(F) - Y)^2 + 2(F - \eta(F))(\eta(F) - Y).$$

Taking expectations, the cross term vanishes since

$$\mathbb{E}[(\eta(F) - Y) \mid F] = 0,$$

and therefore

$$\mathbb{E}[(F - \eta(F))(\eta(F) - Y)] = 0.$$

Moreover, since $Y \mid F \sim \text{Bernoulli}(\eta(F))$,

$$\mathbb{E}\big[(\eta(F) - Y)^2 \mid F\big] = \eta(F)(1 - \eta(F)).$$

## D. Functional Conformal Prediction

We follow (Gu et al., 2025; Xu et al., 2025; Diquigiovanni et al., 2021) for one-sided CP band calculations to determine the time varying threshold $\delta_t$ and use two calibration datasets $\mathcal{D}_{\text{cal}_A}$, $\mathcal{D}_{\text{cal}_B}$. Under the exchangeability assumption (Vovk et al., 2005) and given a user-specified significance level $\alpha$, the CP band guarantees that, for a new successful rollout, its failure probability $1 - f_\theta$ will lie within band $\delta_t$ for all times $t$ with probability $1 - \alpha$. Conversely, if the failure probability of a test rollout rises above that threshold, we can label it as a failure with nominal confidence of $1 - \alpha$. See Alg. 2 for the detailed calculations. Note that this assumption holds for every time step $t$ since we consider the *maximal* failure detection score until time step $t$ and not $1 - f_\theta(h_t)$.

Intuitively, Eq. (9) in Algorithm 2 removes the top $\alpha$ most extreme rollouts from set $\mathcal{D}_{\text{cal}_A}$ before calculating the width of the prediction bands. This ensures the actual band width represents the bulk of the data, not the outliers. Additionally, the maximum over $t$ is taken because the conformal prediction band is intended to reflect the entire trajectory. We define

$$\mathcal{D} = \{D_j \mid j = 1, \ldots, N_2\}$$

as the collection of such maximum deviations from the mean. The band width $h$ is computed as the $(1 - \alpha)$-quantile of $\mathcal{S}$, and the upper bound is given by

$$\tau_t = \text{upper}_t = \mu_t + h \cdot \text{s}_{\text{cal}A}(t)$$

Note that $D_j$'s were dividing by $\text{s}_{\text{cal}A}(t)$ in Eq. (9), we then multiply by this term in line 24 to map the results back to the original scale.

## E. Algorithms

Algorithms 3 and 4 describe the pseudo code for both Early Stopping with Conformal Prediction and Test-Time Guided Action Search.

## F. Experiments Details

### F.1. Code

You can find the Github repository here: https://github.com/shellytechnion/TDQC This repository contains the implementation for this paper. The codebase is built upon and extends the Gu et al. (2025) work on failure detection in Vision-Language-Action (VLA) models.

---

**Algorithm 2** Split-Conformal Trajectory Upper Band with Modulation

---

**input** Calibration rollouts $\mathcal{D}_{\text{cal}}$, miscoverage level $\alpha \in (0, 1)$, scoring network $\mathcal{S}(\cdot; \theta)$, rollout horizon $T$

1: Split $\mathcal{D}_{\text{cal}}$ into two disjoint sets: $\mathcal{D}_{\text{cal}_A} = \{P^k\}_{k=1}^{N_1}$ and $\mathcal{D}_{\text{cal}_B} = \{P^j\}_{j=1}^{N_2}$.

2: **Compute mean successful trajectory on $\mathcal{D}_{\text{cal}_A}$:**

3: **for** $t = 1, \ldots, T$ **do**

4: $\quad \mu_t \leftarrow \frac{1}{N_1} \sum_{k=1}^{N_1} \mathcal{S}(P_t^k; \theta)$

5: **end for**

6: **Compute trimming threshold $\gamma$ on $\mathcal{D}_{\text{cal}_A}$:**

7: **for** $k = 1, \ldots, N_1$ **do**

8: $\quad M_k \leftarrow \max_{t \in [T]} \left| \mathcal{S}(P_t^k; \theta) - \mu_t \right|$

9: **end for**

10: $\gamma \leftarrow \text{Quantile}_{1-\alpha} \left( \{M_k\}_{k=1}^{N_1} \right)$

11: **Define index set $\mathcal{H}$ for modulation computation:**

$$
\mathcal{H} = \begin{cases} [N_1], & \text{if } (N_1 + 1)(1 - \alpha) > N_1, \\ \left\{ k \in [N_1] : \max_{t \in [T]} \left| \mathcal{S}(P_t^k; \theta) - \mu_t \right| \le \gamma \right\}, & \text{otherwise,} \end{cases}
$$

12: **Compute modulation function $s_{\text{cal}_A}(t)$:**

13: **for** $t = 1, \ldots, T$ **do**

14:

$$
s_{\text{cal}_A}(t) \leftarrow \max_{k \in \mathcal{H}} \left| \mathcal{S}(P_t^k; \theta) - \mu_t \right| \tag{9}
$$

15: **end for**

16: **Compute normalized max-deviation scores on $\mathcal{D}_{\text{cal}_B}$:**

17: **for** $j = 1, \ldots, N_2$ **do**

18: $\quad D_j \leftarrow \max_{t \in [T]} \left\{ \frac{\mu_t - \mathcal{S}(P_t^j; \theta)}{s_{\text{cal}_A}(t)} \right\}$

19: **end for**

20: $\mathcal{D} \leftarrow \{D_j \mid j = 1, \ldots, N_2\}$

21: **Compute conformal band width and upper bound:**

22: $h \leftarrow \text{Quantile}_{1-\alpha} (\mathcal{D})$

23: **for** $t = 1, \ldots, T$ **do**

24: $\quad \delta_t \leftarrow \mu_t + h \cdot s_{\text{cal}_A}(t)$

25: **end for**

---

## F.2. VLA models

We show experiments on 4 state-of-the-art open source VLA models: OpenVLA (Kim et al., 2024), UniVLA (Wang et al., 2025), $\pi_0$-FAST (Pertsch et al., 2025) and $\pi_0$ (Black et al., 2024). OpenVLA use MIT license; $\pi_0$ and $\pi_0$-FAST use Apache-2.0 license. At the time of writing, UniVLA does not specify an open-source license in its repository.

## F.3. Action and Token Probability Modeling

**OpenVLA** (Kim et al., 2024) represents an action with $D$ discrete tokens, where each token represents one dimension of the robot's action space (DoF). For each dimension $d \in \{1, \ldots, D\}$, the policy outputs logits $z_t^{(d)}$ and probabilities

$$
p_t^{(d)} = \text{softmax}\left( z_t^{(d)} \right), \qquad p_{t,k}^{(d)} = \pi_\theta \left( A_t^{(d)} = k \mid o_t \right),
$$

for action tokens $k \in \{1, \ldots, K\}$ in an action vocabulary of size $K$. At time $t$, the policy selects the top tokens

$$
a_t^{(d)} = \arg\max_k p_{t,k}^{(d)}
$$

for each action dimension $d$, and decodes them into a continuous action for execution by the robot.

Both the **UniVLA** and $\pi_0$-**FAST** models use the FAST tokenizer (Pertsch et al., 2025), which applies a Discrete Cosine Transform (DCT) to encode continuous action trajectories into discrete action tokens. Given a window of size $H$, we define

---

**Algorithm 3** Early Stopping

---

**input** Policy $\pi$, dataset $\mathcal{D}_{\text{cal}}$, confidence $\alpha$
1: $f_\theta \leftarrow \text{TDQC}(\pi, \mathcal{D}_{\text{cal}})$
2: Sample $\mathcal{D}_{\text{val}}$ i.i.d. from $\mathcal{D}_{\text{cal}}$
3: $\{\delta_t\}_{t=1}^T \leftarrow \text{ConfPred}(\pi, f_\theta, \mathcal{D}_{\text{val}})$ {Algorithm 2}
4: **for** $t = 1, \ldots, T$ **do**
5:     Execute action $a \sim \pi(x_t)$
6:     Append $(a_t, r_t, x_{t+1})$ to $h_{t-1}$
7:     **if** $1 - f_\theta(h_t) > \delta_t$ **then** break
8: **end for**

---

**Algorithm 4** Q-Value Guided Action Search

---

**input** observation $x_t$, policy $\pi$, learned Q-function $f_\theta$, simulator $W$, sample size $M$, threshold $\bar{T}$
1: $a_t = \arg\max_{a'} \pi(a'|x_t)$ {Greedy action}
2: **if** $f_\theta(x_t, a_t) \leq \bar{T}$ **then** {High success confidence}
3:     Execute $a_t$
4: **else**
5:     **for** $i = 1, \ldots, M$ **do**
6:         $a_t^{(i)} \sim \pi(x_t)$ {Sample candidate action}
7:         $x_{t+1}^{(i)} \sim W(x_t, a_t^{(i)})$
8:         $a_{t+1}^{(i)} = \arg\max_{a'} \pi(a'|x_{t+1}^{(i)})$ {Greedy action}
9:         $Q^{(i)} \leftarrow f_\theta(x_{t+1}^{(i)}, a_{t+1}^{(i)})$
10:     **end for**
11:     $i^\star \leftarrow \arg\max_i Q^{(i)}$
12:     Execute $a_t^{(i^\star)}$
13: **end if**

---

the raw continuous action sequence as $A_{1:H} = \{a_1, a_2, \ldots, a_H\}$, where each step $a_t$ is a $d$-dimensional vector. During training, the FAST tokenizer encodes this continuous sequence into a discrete token sequence $L_a = [T_1, \ldots, T_n]$, where each token $T_i$ is drawn from a vocabulary $\mathcal{V}$ of size $|\mathcal{V}|$. Analogous to natural language processing, sequence lengths can vary, resulting in a variable-length $(n)$ discrete representation. During inference, these discrete tokens are decoded back into continuous robot actions $A_{1:H}$. For each token $T_i$, the model outputs logits $z_t^{(T_i)}$ over the vocabulary, which are converted into probabilities via a softmax operation:

$$p_t^{(T_i)} = \text{softmax}\left(z_t^{(T_i)}\right) \tag{10}$$

We refer to these as token probabilities, which we utilize in our "black-box" configuration as $h_t$.

### F.4. Benchmarks Statistics

| Benchmark | Number of Tasks | | | Number of rollouts | | | |
|---|---|---|---|---|---|---|---|
| | Seen | Unseen | Total | Train | Eval Seen | Eval Unseen | Total |
| LIBERO | 7 | 3 | 10 | 210 | 140 | 150 | 500 |
| Real Franka | 10 | 3 | 13 | 450 | 150 | 180 | 780 |
| Real WidowX | 6 | 2 | 8 | 250 | 133 | 149 | 532 |

*Table 3.* Benchmark statistics: task split into seen/unseen subsets and corresponding numbers of training and evaluation rollouts.

Table 3 summarize each benchmark statistics on the number of tasks and rollouts. We note that while TDQC methods requires training data, they exhibit an improvement over the baselines in the real-robot experiments where they are trained on a much smaller dataset. Table 4 shows the number of timesteps per benchmark and VLA model pairs.

| Benchmark | VLA Model | Number of Environment Steps |
|---|---|---|
| LIBERO-10 | OpenVLA | $199,279$ |
| | $\pi_0$ | $151,755$ |
| | $\pi_0$-FAST | $183,488$ |
| | UniVLA | $134,746$ |
| WidowX | OpenVLA | $26,600$ |
| Franka | $\pi_0$-FAST | $36,180$ |

*Table 4.* **Episode horizon by benchmark and model.** We report the maximum rollout length (environment time limit) used in each evaluation setting.

| Model | LIBERO-10 |
|---|---|
| OpenVLA | 52% |
| $\pi_0$-FAST | 60.2% |
| $\pi_0$ | 82.2% |
| UniVLA | 90.6% |

*Table 5.* LIBERO-10 Task success rates across models

**LIBERO-10**   We evaluate the VLA models on LIBERO-10, which consists of 10 long-horizon manipulation tasks and contains the most diverse objects, environments and instructions among the 4 LIBERO suits, thus considered the most challenging suite. We kept the same initial conditions of each enviorment as specified in the benchmark details.[3] We constructed a task-level split into training, validation, and testing: we train and validate on 7 tasks and evaluate generalization on the remaining 3 unseen tasks. The specific tasks split is re-sampled over random seeds, i.e., each seed induces a different set of test tasks. Within the 7 training tasks, we further split rollouts into train/validation with a $60\%/40\%$ ratio. Each task includes 50 randomized initializations (e.g., variations in object placement), meaning a total of $10 \times 50 = 500$ episodes. Table 5 reports the rollout success rates of all evaluated VLA policies.

In LIBERO, the simulator terminates an episode as soon as the task succeeds, or when a maximum horizon is reached (520 steps for LIBERO-10). As a result, successful rollouts are often shorter than failed ones, which can cause a success estimators to induce information based on the length of the rollout alone. To ensure a fair comparison of ROC-AUC, we follow a fixed-cutoff method that was introduced in Gu et al. (2025). For each task, we compute the minimum rollout length observed across its episodes and use this value as the effective horizon $T$ for evaluation $T_k = \min_{r \in k} T_r$ where $k$ denotes the task and $r$ is a rollout. Meaning, we only report ROC-AUC values up to this termination time. This ensures that all rollouts for a given task are evaluated on comparable trajectory prefixes.

**Real Robot WidowX**   We used published data[4] that tested OpenVLA pretrained on "Open-X Magic soup++" dataset on real WidowX robot arm which achieved success rate of $45.8\%$. In this experiment, they collected 532 rollouts on the 8 tasks, with 244 successes and 288 failures. Each task has roughly the same number of rollouts. We constructed a task-level split into training, validation, and testing: we train and validate on 6 tasks and evaluate generalization on the remaining 2 unseen tasks. The specific tasks split is re-sampled over random seeds, i.e., each seed induces a different set of test tasks. When we evaluated on multiple seeds, each seed determines a different configuration of train and test tasks split. Within the 6 training tasks, we further split rollouts into train/validation with a $66\%/33\%$ ratio. For each task, they set a fixed number of allowed time steps (50) and all rollouts ended in the same time step, even if success occurred before that time step.

**Real-world Franka**   We used published data[5] that tested $\pi_0$-FAST on various real-world experiments. For each task, the maximum time horizon was predetermined and constant inside each task. We split to train, validation and test such that we train and validate on 10 tasks and test on 3 tasks. The authors of (Gu et al., 2025) recorded the rollouts such that the dataset will have 50% success on each task.

---

[3]https://github.com/Lifelong-Robot-Learning/LIBERO/tree/master/libero/libero/init_files/libero_10

[4]https://github.com/vla-safe/SAFE

[5]https://github.com/vla-safe/SAFE

## F.5. Static baselines

Many VLA models discretize continuous actions into descrete action tokens. At time step $t$, the policy produces a distribution over tokens conditioned on the trajectory history and the language instruction,

$$p_t(a) = \pi_\theta(a \mid h_t, l), \qquad a_t = \arg\max_a p_t(a), \tag{11}$$

where $h_t$ denotes the available history (e.g., past observations and actions) and $l$ is the language instruction. For robots with $|\mathcal{D}|$ degrees of freedom (DoF), we write the predicted token vector as $A_t = (a_t^1, \ldots, a_t^{|\mathcal{D}|})$, with corresponding probabilities $P_t = (p_t^1, \ldots, p_t^{|\mathcal{D}|})$, where $p_t^i \triangleq p_t(a_t^i) \in (0, 1]$ is the probability assigned to the selected token for DoF $i$. Let $H_t^i$ denote the entropy of the token distribution for DoF $i$ at time $t$.

We follow the LLM hallucinations literature (Xiong et al., 2023; Huang et al., 2024; Shorinwa et al., 2025; Ren et al., 2023) and Zollo & Zemel (2025) and used the same methods to measure uncertainty in VLA model. We compute the following uncertainty scores at each timestep. For probability-based scores we use the negative log-probability, so that larger values indicate higher uncertainty:

$$\text{Max Prob}(t) = \max_i(-log(p_t^i))$$

$$\text{Avg Prob}(t) = -\frac{1}{|\mathcal{D}|} \sum_{i=1}^{|\mathcal{D}|} log(p_t^i)$$

$$\text{Running Avg Prob}(t) = -\frac{1}{t}\frac{1}{|\mathcal{D}|} \sum_{j=1}^{t} \sum_{i=1}^{|\mathcal{D}|} log(p_j^i)$$

$$\text{Avg Entropy}(t) = \frac{1}{|\mathcal{D}|} \sum_{i=1}^{|\mathcal{D}|} H_t^i$$

$$\text{Running Avg Entropy}(t) = \frac{1}{t}\frac{1}{|\mathcal{D}|} \sum_{j=1}^{t} \sum_{i=1}^{|\mathcal{D}|} H_j^i$$

The first two families use the confidence of the *chosen* action's DoF (via $-\log p_t^i$), i.e. the token in the action that corresponds to the highest probability, while the entropy scores uses the full token distributions. Finally, Running Avg Prob aggregates uncertainty over time via a running average, which can be interpreted as a lightweight temporal smoothing baseline (without learning).

## F.6. Trained Baselines

**SAFE-RNN** uses an LSTM model with 1 layer and a hidden dimension of 256, and an additional linear layer is used to project the hidden states of LSTM into a single scalar state. **SAFE-MLP** uses a multi-layer perceptron with 2 layers and a hidden dimension of 256. RNN-BCE and RNN-TDQC methods use the same LSTM architecture for all benchmarks, except OpenVLA, where the LSTM is replaced by a GRU. In **OpenVLA with action probabilities**, we trained a GRU network with A two-layer MLP with and a 1 layer GRU, the GRU's output is passed through a two-layer MLP head that produces a single scalar output. Since successes and failures from the generated rollouts are imbalanced, the losses on positive (failed) and negative (successful) rollouts are weighted by their inverse class frequency.

Given the internal feature vectors $E \in \mathbb{R}^{n \times d}$ produced by a VLA model, where $n$ corresponds to token positions, diffusion steps, etc. and $d$ is the feature dimension. Gu et al. (2025) aggregate $E$ into a single fixed-dimensional feature vector $e \in \mathbb{R}^d$ before inputting to SAFE models. They ablate on which feature vector to use in order to achieve highest ROC-AUC. In $\pi_0$ Gu et al. (2025) takes the feature vectors before the projection into the velocity field. The aggregation methods in $\pi_0$ are denoted as $agg_{\text{hori}}$, $agg_{\text{hdiff}}$ Other implementation details appear both in SAFE paper (Gu et al., 2025) and in https://github.com/vla-safe/SAFE.

### F.7. Training Details

Following SAFE, for both metrics sequential Brier Score and ROC-AUC, we follow the evaluation in SAFE and interpreted the uncertainty estimations (probabilities) as the *probability of failure*.

We use AdamW optimizer (Loshchilov & Hutter, 2017) with weight decay of 0.01 , and learning rate (lr) that was determined by grid search and a step LR scheduler with gamma determined by a grid search. All models were trained for 1000 epochs with batch size 512. We also ablate on applying L2 regularization $\lambda_{\text{reg}}$ loss on the model weights to reduce weights. Note that for the TDQC method that gets as an input the model's probabilities we entered each data point sequentially. We concatenated all the rollount and samples "starting index" from which we took a batch size next steps. If the rollout ended in the middle of the batch - we reset the network's hidden state. In SAFE methods, each rollout is considered as one data point and thus batch size of 512 translates to training on (at most) 512 rollouts in each iteration. All training and evaluation are done on a single NVIDIA RTX 5090 32GB GPU.

### F.8. Hyperparameter Tuning

To tune TDQC, we perform a grid search over hyperparameters and select the configuration that **maximizes sequential Brier score** on a held-out validation set of rollouts from the same tasks as in training. Unlike Gu et al. (2025), which evaluates each hyperparameter setting across all random seeds and then reports the best-performing one (in terms of ROC-AUC), we decouple tuning and evaluation: we run the hyperparameter sweep on two tuning seeds and then fix the selected configuration and evaluate it on an additional nineteen seeds (21 total). Using a large number of evaluation seeds reduces selection bias and yields statistically meaningful comparisons.

In Tables 6 to 11 we report the hyper-parameters we have searched over and the values of the best performance. For the SAFE methods - we used their chosen parameters for the LSTM and MLP methods. For MLP-BCE method we searched the same grid to find the best parameters. For SAFE methods without TD we choose the parameters that maximized the ROC-AUC, and for the TDQC parameters, we choose the ones that minimize the sequential Brier Score. Unless stated otherwise, in action probability methods (OpenVLA model) we concatenated the top 10 probabilities for each degree of freedom, used learning rate step size of 200 and batch size of 512. For all methods, $\text{LR}_\lambda = 1$, and the number of layers is 2 for the projection in MLP and for LSTM method we used 1 LSTM layer.

*Table 6.* Hyperparameter search space for OpenVLA + LIBERO benchmark.

| Method | HParams | Values |
|---|---|---|
| *SAFE-RNN* | $agg_{\text{token}}$ | First **Last** Mean |
| | lr | **1e-4** 3e-4 1e-3 |
| | $\lambda_{\text{reg}}$ | 1e-3 1e-2 1e-1 **1** |
| *SAFE-RNN-TDQC* | $agg_{\text{token}}$ | First **Last** Mean |
| | lr | 1e-5 5e-5 **1e-4** 3e-4 1e-3 |
| | lr $\gamma$ | 0.1 **0.8** |
| | $\lambda_{\text{reg}}$ | **0** 1e-3 1e-1 |
| *SAFE-MLP* | $agg_{\text{token}}$ | First **Last** Mean |
| | lr | **1e-4** 3e-4 1e-3 |
| | $\lambda_{\text{reg}}$ | 1e-3 **1e-2** 1e-1 1 |
| *SAFE-MLP BCE* | $agg_{\text{token}}$ | First **Last** Mean |
| | lr | 1e-4 **3e-4** 1e-3 |
| | $\lambda_{\text{reg}}$ | 1e-3 1e-2 **1e-1** 1 |
| *SAFE-MLP-TDQC* | $agg_{\text{token}}$ | First **Last** Mean |
| | lr | 1e-5 5e-5 **1e-4** 3e-4 1e-3 |
| | $\lambda_{\text{reg}}$ | **0** 1e-3 1e-1 |
| | LR-$\gamma$ | **0.8** 0.1 |
| *RNN-BCE* | GRU hidden | 256 **512** |
| | head hidden | 256 **512** |
| | lr | **1e-5** 5e-5 1e-4 1e-3 |
| | lr $\gamma$ | 0.1 **0.8** |
| | $\lambda_{\text{reg}}$ | **0** 1e-3 1e-2 1e-1 |
| *RNN-TDQC* | GRU hidden | **256** 512 |
| | head hidden | 256 **512** |
| | lr | **1e-5** 5e-5 1e-4 1e-3 |
| | lr $\gamma$ | 0.1 **0.8** |
| | $\lambda_{\text{reg}}$ | **0** 1e-3 1e-2 1e-1 |

*Table 7.* Hyperparameter search space for OpenVLA + WidowX benchmark.

| Method | HParams | Values |
|---|---|---|
| *SAFE-RNN* | $agg_{\text{token}}$ | First **Last** Mean |
| | lr | 1e-4 **3e-4** 1e-3 |
| | $\lambda_{\text{reg}}$ | 1e-3 **1e-2** 1e-1 **1** |
| *SAFE-RNN-TDQC* | $agg_{\text{token}}$ | First **Last** Mean |
| | lr | 1e-5 **5e-5** 1e-4 3e-4 1e-3 |
| | lr $\gamma$ | **0.1** 0.8 |
| | $\lambda_{\text{reg}}$ | 0 **1e-3** 1e-1 |
| *SAFE-MLP* | $agg_{\text{token}}$ | First **Last** Mean |
| | lr | 1e-4 **3e-4** 1e-3 |
| | $\lambda_{\text{reg}}$ | 1e-3 1e-2 **1e-1** 1 |
| *SAFE-MLP BCE* | $agg_{\text{token}}$ | First **Last** Mean |
| | lr | **1e-5** 5e-5 1e-4 3e-4 1e-3 |
| | $\lambda_{\text{reg}}$ | 0 1e-3 **1e-1** 1 |
| *SAFE-MLP-TDQC* | $agg_{\text{token}}$ | First **Last** Mean |
| | lr | 1e-5 5e-5 1e-4 **3e-4** 1e-3 |
| | $\lambda_{\text{reg}}$ | **0** 1e-3 1e-1 |
| | LR-$\gamma$ | **0.1** 0.8 |
| *RNN-BCE* | GRU hidden | **256** 512 1024 |
| | head hidden | 256 **512** |
| | lr | 1e-5 5e-5 1e-4 3e-4 **1e-3** |
| | lr $\gamma$ | **0.1** 0.8 |
| | $\lambda_{\text{reg}}$ | 0 **1e-3** 1e-1 |
| *RNN-TDQC* | GRU hidden | 256 512 **1024** |
| | head hidden | 256 **512** |
| | lr | **1e-5** 5e-5 1e-4 3e-4 1e-3 |
| | lr $\gamma$ | **0.1** 0.8 |
| | $\lambda_{\text{reg}}$ | **0** 1e-3 1e-1 |

*Table 8.* Hyperparameter search space for $\pi_0$-FAST + LIBERO benchmark.

| Method | HParams | Values |
|---|---|---|
| SAFE-RNN | $agg_{\text{token}}$ | First Last **Mean** |
| | Feat | **Encoded** Pre-logits |
| | lr | 3e-5 1e-4 **3e-4** 1e-3 |
| | $\lambda_{\text{reg}}$ | **1e-3** 1e-2 1e-1 |
| SAFE-RNN-TDQC | $agg_{\text{token}}$ | First Last **Mean** |
| | Feat | Encoded **Pre-logits** |
| | lr | 1e-5 5e-5 1e-4 3e-4 **1e-3** |
| | lr $\gamma$ | **0.1** 0.8 |
| | $\lambda_{\text{reg}}$ | **0** 1e-3 1e-1 |
| SAFE-MLP | $agg_{\text{token}}$ | First **Last** Mean |
| | Feat | Encoded **Pre-logits** |
| | lr | 3e-5 **1e-4** 1e-4 1e-3 |
| | $\lambda_{\text{reg}}$ | 1e-3 **1e-2** 1e-1 |
| SAFE-MLP BCE | $agg_{\text{token}}$ | First Last **Mean** |
| | Feat | Encoded **Pre-logits** |
| | lr $\gamma$ | 0.1 **0.8** |
| | lr | 1e-5 5e-5 **1e-4** 3e-4 1e-3 |
| | $\lambda_{\text{reg}}$ | 0 **1e-3** 1e-1 |
| SAFE-MLP-TDQC | $agg_{\text{token}}$ | First Last **Mean** |
| | Feat | Encoded **Pre-logits** |
| | lr | 1e-5 5e-5 1e-4 3e-4 **1e-3** |
| | $\lambda_{\text{reg}}$ | **0** 1e-3 1e-1 |
| | lr $\gamma$ | 0.1 **0.8** |
| RNN-BCE | lr | 1e-5 3e-5 1e-4 3e-4 **1e-3** |
| | $\lambda_{\text{reg}}$ | 1e-3 **1e-1** 0 |
| | lr $\gamma$ | 0.1 **0.8** |
| RNN-TDQC | lr | **1e-5** 3e-5 1e-4 3e-4 1e-3 |
| | $\lambda_{\text{reg}}$ | 1e-3 1e-1 **0** |
| | lr $\gamma$ | **0.1** 0.8 |

*Table 9.* Hyperparameter search space for $\pi_0$-FAST + Franka benchmark.

| Method | HParams | Values |
|---|---|---|
| SAFE-RNN | $agg_{\text{token}}$ | **Mean** |
| | Feat | **Pre-logits** |
| | hidden dim | 128 **256** |
| | lr | 1e-4 3e-4 **1e-3** 3e-3 |
| | $\lambda_{\text{reg}}$ | 1e-3 **1e-2** 1e-1 |
| SAFE-RNN-TDQC | $agg_{\text{token}}$ | **Mean** |
| | Feat | **Pre-logits** |
| | hidden dim | 128 **256** |
| | lr | 1e-5 5e-5 1e-4 3e-4 **1e-3** |
| | lr $\gamma$ | 0.1**0.8** |
| | $\lambda_{\text{reg}}$ | **0** 1e-3 1e-1 |
| SAFE-MLP | $agg_{\text{token}}$ | **Mean** |
| | Feat | **Pre-logits** |
| | lr | 1e-4 **3e-4** 1e-3 3e-3 |
| | $\lambda_{\text{reg}}$ | **1e-3** 1e-2 1e-1 |
| SAFE-MLP BCE | $agg_{\text{token}}$ | **First** Last Mean |
| | Feat | **Pre-logits** |
| | hidden dim | **128** 256 |
| | lr $\gamma$ | 0.1 **0.8** |
| | lr | 1e-5 5e-5 1e-4 **3e-4** 1e-3 |
| | $\lambda_{\text{reg}}$ | **0** 1e-3 1e-1 |
| SAFE-MLP-TDQC | $agg_{\text{token}}$ | **Mean** |
| | Feat | **Pre-logits** |
| | hidden dim | 128 **256** |
| | lr | 1e-5 5e-5 1e-4 3e-4 **1e-3** |
| | $\lambda_{\text{reg}}$ | **0** 1e-3 1e-1 |
| | lr $\gamma$ | 0.1 **0.8** |
| RNN-BCE | hidden dim | 128 **256** |
| | lr | **1e-5** 3e-5 1e-4 3e-4 1e-3 |
| | $\lambda_{\text{reg}}$ | 1e-3 **1e-1** 0 |
| | lr $\gamma$ | **0.1** 0.8 |
| RNN-TDQC | hidden dim | 128 **256** |
| | lr | 1e-5 3e-5 **1e-4** 3e-4 1e-3 |
| | $\lambda_{\text{reg}}$ | **1e-3** 1e-1 0 |
| | lr $\gamma$ | **0.1** 0.8 |

*Table 10.* Hyperparameter search space for $\pi_0$ + LIBERO benchmark.

| Method | HParams | Values |
|---|---|---|
| SAFE-RNN | $agg_{\text{horri}}$ | **First** Last First&Last |
| | $agg_{\text{diff}}$ | First **Last** First&Last |
| | lr | 1e-5 3e-5 1e-4 3e-4 **1e-3** |
| | $\lambda_{\text{reg}}$ | **1e-3** 1e-2 1e-1 |
| SAFE-RNN-TDQC | $agg_{\text{horri}}$ | **First** Last First&Last |
| | $agg_{\text{diff}}$ | First Last **First&Last** |
| | lr | 1e-5 3e-5 1e-4 **3e-4** 1e-3 |
| | lr $\gamma$ | **0.1** 0.8 |
| | $\lambda_{\text{reg}}$ | **0** 1e-3 1e-1 |
| SAFE-MLP | $agg_{\text{horri}}$ | **First** Last First&Last |
| | $agg_{\text{diff}}$ | First **Last** First&Last |
| | lr | 1e-5 **3e-5** 1e-4 3e-4 1e-3 |
| | $\lambda_{\text{reg}}$ | **1e-3** 1e-2 1e-1 |
| SAFE-MLP BCE | $agg_{\text{horri}}$ | First Last **First&Last** |
| | $agg_{\text{diff}}$ | First **Last** First&Last |
| | lr | 1e-5 **3e-5** 1e-4 3e-4 1e-3 |
| | lr $\gamma$ | 0.1 **0.8** |
| | $\lambda_{\text{reg}}$ | **0** 1e-3 1e-1 |
| SAFE-MLP-TDQC | $agg_{\text{horri}}$ | First Last **First&Last** |
| | $agg_{\text{diff}}$ | First **Last** First&Last |
| | lr | 1e-5 3e-5 1e-4 3e-4 **1e-3** |
| | lr $\gamma$ | 0.1 **0.8** |
| | $\lambda_{\text{reg}}$ | **0** 1e-3 1e-1 |

*Table 11.* Hyperparameter search space for UniVLA + LIBERO benchmark.

| Method | HParams | Values |
|---|---|---|
| SAFE-RNN | $agg_{\text{token}}$ | **First** Last Mean |
| | lr | **1e-5** 3e-5 1e-4 3e-4 1e-3 |
| | $\lambda_{\text{reg}}$ | 1e-3 **1e-1** 0 |
| | lr $\gamma$ | **0.1** 0.8 |
| SAFE-RNN-TDQC | $agg_{\text{token}}$ | **First** Last Mean |
| | lr | 1e-5 **3e-5** 1e-4 3e-4 1e-3 |
| | $\lambda_{\text{reg}}$ | **1e-3** 1e-1 0 |
| | lr $\gamma$ | **0.1** 0.8 |
| SAFE-MLP | $agg_{\text{token}}$ | **First** Last Mean |
| | lr | 1e-5 3e-5 1e-4 3e-4 **1e-3** |
| | $\lambda_{\text{reg}}$ | 1e-3 **1e-1** 0 |
| | lr $\gamma$ | **0.1** 0.8 |
| SAFE-MLP BCE | $agg_{\text{token}}$ | **First** Last Mean |
| | lr | **1e-5** 3e-5 1e-4 3e-4 1e-3 |
| | $\lambda_{\text{reg}}$ | 1e-3 **1e-1** 0 |
| | lr $\gamma$ | **0.1** 0.8 |
| SAFE-MLP-TDQC | $agg_{\text{token}}$ | **First** Last Mean |
| | lr | 1e-5 3e-5 **1e-4** 3e-4 1e-3 |
| | $\lambda_{\text{reg}}$ | **1e-3** 1e-1 0 |
| | lr $\gamma$ | 0.1 **0.8** |
| RNN-BCE (Top 10) | lr | 1e-5 3e-5 1e-4 3e-4 **1e-3** |
| | $\lambda_{\text{reg}}$ | 1e-3 1e-1 **0** |
| | lr $\gamma$ | **0.1** 0.8 |
| RNN-TDQC | lr | 1e-5 3e-5 **1e-4** 3e-4 1e-3 |
| | $\lambda_{\text{reg}}$ | **1e-3** 1e-1 0 |
| | lr $\gamma$ | 0.1 **0.8** |

# G. Additional Results

## G.1. Results variance

In Table 12 and Table 13 we report the standard deviation for all results in Tables 1 and 2. Note that the reported values were averaged over 21 seeds which set the environment seed and determines the train-test split of tasks. Since different tasks have different difficulties it is reasonable to see large standard deviations in the tables. Tables 12 and 13 shows that across models and benchmarks, *TDQC learning improves calibration relative to non-TDQC variants*, with relatively lower standard deviations compared to the baselines, where SAFE-RNN-TDQC performs the best or on par for all $\pi_0$ model variants.

| VLA Model | OpenVLA | | OpenVLA | | UniVLA | | $\pi_0$-FAST | | $\pi_0$-FAST | | $\pi_0$ | |
| Benchmark | LIBERO | | WidowX | | LIBERO | | LIBERO | | Franka | | LIBERO | |
| Eval Task Split | Seen ↓ | Unseen ↓ | Seen ↓ | Unseen ↓ | Seen ↓ | Unseen ↓ | Seen ↓ | Unseen ↓ | Seen ↓ | Unseen ↓ | Seen ↓ | Unseen ↓ |
|---|---|---|---|---|---|---|---|---|---|---|---|---|
| Max prob. | 0.395 ± 0.037 | 0.390 ± 0.028 | 0.572 ± 0.060 | 0.579 ± 0.012 | 0.909 ± 0.024 | 0.899 ± 0.048 | 0.320 ± 0.021 | 0.318 ± 0.022 | 0.331 ± 0.018 | 0.323 ± 0.017 | – | – |
| Avg prob. | 0.348 ± 0.036 | 0.364 ± 0.061 | 0.275 ± 0.012 | 0.282 ± 0.019 | 0.529 ± 0.017 | 0.532 ± 0.035 | 0.212 ± 0.019 | 0.218 ± 0.024 | 0.290 ± 0.016 | 0.294 ± 0.024 | – | – |
| Running Avg prob. | 0.338 ± 0.034 | 0.356 ± 0.059 | 0.255 ± 0.003 | 0.257 ± 0.002 | 0.543 ± 0.016 | 0.544 ± 0.031 | 0.244 ± 0.024 | 0.238 ± 0.045 | 0.359 ± 0.029 | 0.361 ± 0.026 | – | – |
| Avg entropy | 0.306 ± 0.020 | 0.313 ± 0.030 | 0.414 ± 0.044 | 0.426 ± 0.095 | 0.406 ± 0.027 | 0.391 ± 0.063 | 0.209 ± 0.022 | 0.222 ± 0.033 | 0.281 ± 0.017 | 0.281 ± 0.021 | – | – |
| Running Avg entropy | 0.265 ± 0.014 | 0.273 ± 0.026 | 0.435 ± 0.028 | 0.432 ± 0.054 | 0.343 ± 0.016 | 0.330 ± 0.039 | 0.279 ± 0.021 | 0.264 ± 0.046 | 0.341 ± 0.025 | 0.339 ± 0.027 | – | – |
| *SAFE-RNN* | 0.204 ± 0.038 | 0.255 ± 0.059 | 0.169 ± 0.061 | 0.213 ± 0.075 | 0.124 ± 0.020 | 0.162 ± 0.014 | 0.106 ± 0.018 | 0.148 ± 0.042 | 0.220 ± 0.047 | 0.288 ± 0.055 | 0.123 ± 0.043 | 0.172 ± 0.095 |
| *SAFE-RNN-TDQC (Ours)* | 0.197 ± 0.024 | 0.218 ± 0.020 | 0.096 ± 0.021 | 0.153 ± 0.056 | 0.064 ± 0.016 | 0.100 ± 0.026 | 0.103 ± 0.028 | 0.163 ± 0.060 | 0.150 ± 0.026 | 0.215 ± 0.038 | 0.061 ± 0.019 | 0.097 ± 0.033 |
| *SAFE-MLP BCE* | 0.192 ± 0.020 | 0.231 ± 0.020 | 0.127 ± 0.019 | 0.164 ± 0.034 | 0.091 ± 0.022 | 0.158 ± 0.036 | 0.103 ± 0.019 | 0.162 ± 0.053 | 0.206 ± 0.016 | 0.248 ± 0.023 | 0.075 ± 0.018 | 0.137 ± 0.058 |
| *SAFE-MLP-TDQC (Ours)* | 0.195 ± 0.022 | 0.229 ± 0.022 | 0.130 ± 0.022 | 0.169 ± 0.038 | 0.066 ± 0.015 | 0.131 ± 0.028 | 0.109 ± 0.022 | 0.150 ± 0.035 | 0.210 ± 0.008 | 0.229 ± 0.013 | 0.068 ± 0.021 | 0.128 ± 0.060 |
| *RNN-BCE* | 0.199 ± 0.016 | 0.206 ± 0.022 | 0.301 ± 0.062 | 0.344 ± 0.048 | 0.138 ± 0.047 | 0.152 ± 0.065 | 0.122 ± 0.022 | 0.158 ± 0.053 | 0.238 ± 0.006 | 0.243 ± 0.005 | – | – |
| *RNN-TDQC (Ours)* | 0.191 ± 0.015 | 0.197 ± 0.021 | 0.156 ± 0.023 | 0.192 ± 0.039 | 0.100 ± 0.025 | 0.107 ± 0.056 | 0.105 ± 0.020 | 0.141 ± 0.045 | 0.205 ± 0.015 | 0.228 ± 0.022 | – | – |

*Table 12.* Brier Score results on simulation and real robot experiment (lower is better). Results are averaged over 21 seeds that determined different train-test split of tasks. "−" indicates that the Brier score can't be calculated on the method. The **first** and second best performing methods are highlighted in the table.

| VLA Model | OpenVLA | | OpenVLA | | UniVLA | | $\pi_0$-FAST | | $\pi_0$-FAST | | $\pi_0$ | |
| Benchmark | LIBERO | | WidowX | | LIBERO | | LIBERO | | Franka | | LIBERO | |
| Eval Task Split | Seen ↑ | Unseen ↑ | Seen ↑ | Unseen ↑ | Seen ↑ | Unseen ↑ | Seen ↑ | Unseen ↑ | Seen ↑ | Unseen ↑ | Seen ↑ | Unseen ↑ |
|---|---|---|---|---|---|---|---|---|---|---|---|---|
| Max prob. | 54.64 ± 5.36 | 55.78 ± 4.33 | 53.25 ± 3.36 | 53.20 ± 2.70 | 50.00 ± 0.00 | 50.00 ± 0.00 | 61.75 ± 8.85 | 63.49 ± 13.6 | 48.61 ± 4.58 | 46.64 ± 6.95 | – | – |
| Avg prob. | 47.08 ± 4.07 | 48.09 ± 5.14 | 47.47 ± 3.99 | 48.30 ± 4.75 | 42.96 ± 8.89 | 40.25 ± 7.71 | 47.36 ± 7.64 | 48.09 ± 7.52 | 49.45 ± 5.41 | 48.03 ± 8.08 | – | – |
| Running Avg prob. | 49.19 ± 4.16 | 47.72 ± 4.38 | 49.20 ± 4.51 | 46.37 ± 6.82 | 43.05 ± 6.14 | 40.29 ± 7.54 | 53.95 ± 4.42 | 55.68 ± 7.40 | 52.95 ± 3.92 | 49.98 ± 4.19 | – | – |
| Avg entropy | 46.81 ± 4.22 | 46.75 ± 4.77 | 50.19 ± 4.64 | 49.36 ± 8.16 | 41.34 ± 9.42 | 47.36 ± 11.8 | 45.42 ± 7.44 | 46.30 ± 7.22 | 49.28 ± 4.34 | 49.07 ± 7.42 | – | – |
| Running Avg entropy | 50.48 ± 4.34 | 48.09 ± 4.76 | 46.16 ± 5.38 | 43.99 ± 8.08 | 51.63 ± 8.50 | 56.71 ± 7.78 | 53.78 ± 5.27 | 55.44 ± 6.86 | 51.12 ± 4.53 | 48.78 ± 5.02 | – | – |
| *SAFE-RNN* | 72.30 ± 4.73 | 69.04 ± 7.01 | 75.95 ± 7.29 | 70.00 ± 5.95 | 74.48 ± 9.49 | 68.13 ± 11.4 | 91.26 ± 3.18 | 85.88 ± 7.63 | 70.85 ± 5.92 | 56.69 ± 7.33 | 79.95 ± 9.22 | 65.03 ± 20.5 |
| *SAFE-RNN-TDQC (Ours)* | 71.67 ± 4.00 | 65.12 ± 4.94 | 84.01 ± 3.89 | 70.76 ± 7.72 | 73.83 ± 8.52 | 64.25 ± 12.6 | 92.03 ± 3.37 | 85.49 ± 6.96 | 79.89 ± 6.80 | 68.43 ± 7.18 | 88.66 ± 4.54 | 82.94 ± 10.7 |
| *SAFE-MLP* | 73.56 ± 3.85 | 69.53 ± 5.02 | 88.23 ± 4.37 | 83.18 ± 6.36 | 74.39 ± 7.73 | 63.80 ± 13.32 | 79.00 ± 12.5 | 68.36 ± 24.3 | 79.01 ± 4.17 | 63.69 ± 7.08 | 86.33 ± 4.94 | 79.75 ± 13.0 |
| *SAFE-MLP-BCE* | 72.66 ± 3.39 | 64.99 ± 5.26 | 85.38 ± 3.49 | 71.43 ± 8.49 | 78.83 ± 5.86 | 69.74 ± 13.9 | 91.82 ± 3.23 | 85.50 ± 6.58 | 73.82 ± 3.61 | 58.83 ± 7.31 | 89.71 ± 3.36 | 80.53 ± 13.8 |
| *SAFE-MLP-TDQC (Ours)* | 71.22 ± 3.23 | 60.09 ± 7.09 | 82.33 ± 3.71 | 70.64 ± 9.42 | 76.55 ± 5.68 | 66.71 ± 12.6 | 90.26 ± 3.22 | 84.40 ± 7.62 | 61.95 ± 6.47 | 51.22 ± 6.53 | 86.57 ± 4.99 | 72.07 ± 18.4 |
| *RNN-BCE* | 72.70 ± 4.26 | 72.28 ± 4.26 | 69.69 ± 6.29 | 67.09 ± 5.48 | 63.44 ± 7.92 | 54.37 ± 12.5 | 87.67 ± 4.11 | 82.53 ± 10.4 | 59.32 ± 4.55 | 53.26 ± 7.34 | – | – |
| *RNN-TDQC (Ours)* | 74.20 ± 3.70 | 72.72 ± 4.29 | 78.90 ± 7.46 | 72.97 ± 6.69 | 72.30 ± 7.52 | 70.17 ± 11.5 | 87.51 ± 3.65 | 83.39 ± 9.67 | 64.10 ± 4.11 | 57.37 ± 9.30 | – | – |

*Table 13.* ROC-AUC results on simulation and real robot experiment (higher is better). Results are averaged over 21 seeds that determined different train-test split of tasks. "−" indicates that the ROC-AUC can't be calculated on the method. The **first** and second best performing methods are highlighted in the table.

## G.2. Generalization across benchmarks

To assess the generalization capability of the success predictors, we trained an RNN-TDQC (top-10) predictor on OpenVLA trajectories from the LIBERO benchmark and evaluated it on OpenVLA trajectories from the WidowX benchmark. On the unseen WidowX tasks, the predictor achieved a ROC-AUC of $53.69 \pm 5.07$ and a sequential Brier score of $0.413 \pm 0.03$, compared to $72.97 \pm 6.69$ and $0.192 \pm 0.039$, respectively, on the original LIBERO tasks. These results suggest that the success predictor has limited generalization capabilities.

## G.3. Extended Evaluation for Application to Test-Time Action Search

In order to extend out evaluations for test-time action search, we added to Figure 4 ablation configurations: *Probs Entropy* is a heuristic approach that does not rely on a learned Q-value, serving as a natural, zero-shot baseline. The inclusion of this method is motivated by the need to determine whether a separately trained value network $f_\theta$ is necessary; it tests the hypothesis that the base VLA model's internal predictive certainty is a sufficient indicator of action quality. It is the 'Avg Entropy' method from Tables 1 and 2, which achieved good sequential brier and ROC-AUC compared the the static baselines. It selects the action with lowest mean entropy across all $D$ action dimensions:

$$a_t = \arg\min_{a_t} \left\{ \frac{1}{D} \sum_{d=1}^{D} H\left( \text{PDF}^{(d)} \right) \right\}$$

More details are in Apx. F.5.

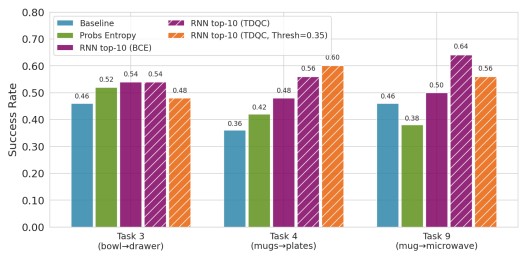

*(a)* Success rate comparison across specific tasks, including ablation configurations.

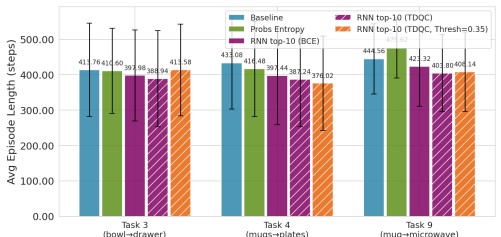

*(b)* Average episode length (steps) per task, including ablation configurations.

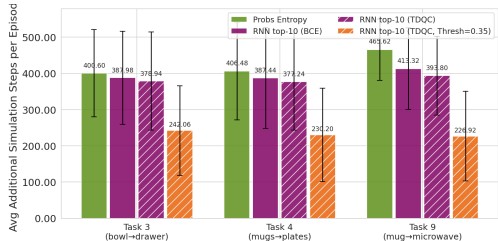

*(c)* Averaged number of steps used action search per task, including ablation configurations.

*Figure 5.* **Extended Analysis of Guided Action Search and TDQC Efficiency.** The results demonstrate that *RNN-TDQC* provides the highest success rates, while the *Threshold 0.35* variant offers a significant reduction in computational overhead by selectively triggering action search only when safety is at risk while maintaining high success rates.

While results in the paper summarized the average success rate across all tasks, we provide a more granular breakdown in Figure 5a-c to analyze per-task performance and computational trade offs. Figure 5a details the success rates for 3 unseen tasks: placing a *bowl in a drawer* (Task 3), moving *two mugs to plates* (Task 4), and placing a *mug in a microwave* (Task 9). We observe that TDQC-based methods consistently outperform the baseline across all unseen tasks, with *RNN-TDQC* achieving the highest success rate across all configurations.

As shown in Figure 5c, while the *TDQC-Threshold 0.35* variant achieves success rates that exceed the baseline, its primary advantage lies in computational efficiency. By comparing the average episode length (Figure 5b) with the actual number of steps triggered the action search (Figure 5c), it is evident that this strategy significantly reduces total compute requirements while maintaining competitive performance.

### G.4. Computational Cost

For real robot deployment, test-time compute for TDQC is not more costly than baselines such as SAFE: both require only a forward pass of the MLP/LSTM at every time step, typically much lighter than the VLA forward pass. Furthermore, the network for TDQC with action probabilities as input is lightweight, and smaller than SAFE with features as input. RL theory suggests that TD-based methods require longer training than Monte-Carlo (MC) methods, thus, we evaluated all methods on the same GPU (NVIDIA RTX 5090) with 5 seeds on various computational metrics. Results are reported in Tables 14 and 15. From our analysis, the overhead from the TD loss is small, 25% on OpenVLA and 15% on $\pi_0$-FAST.

### G.5. quantitative results of RNN-TDQC with top 10 probabilities

We visualize rollout trajectories together with the failure scores predicted by TDQC on OpenVLA LIBERO-10 benchmark. Figure 6 illustrates both failure and success cases. The green-shaded areas show the functional CP band. Once failure scores exceed the band, a failure flag is raised. Figure 6a shows an example of the flag not being raised during a successful rollout, whereas Figure 6b shows that the failure flag is triggered once the policy becomes stuck. Our quantitative analysis further indicates All methods reliably detect failure modes in which the policy gets stuck or oscillates with slow back-and-forth motions. For RNN methods, once the robot successfully grasps an object, the failure score drops, suggesting that the model recognizes this as a 'good' action.
Figure 7 shows a successful rollout with an informative failure score. The failure score rises when the policy becomes

*Table 14.* Computational cost of all failure prediction methods trained on OpenVLA LIBERO-10 benchmark. We report reserved and peak GPU VRAM (MB), number of trainable parameters (M), total training wall-clock time (s), and per-epoch time (s), averaged over five seeds. TD-0 loss adds ~25% overhead relative to BCE for a given architecture.

| run_name | vram reserved mb | num params M | train wall clock sec | peak vram mb | epoch time sec |
|---|---|---|---|---|---|
| MLP (features, BCE Loss) | $12104.8 \pm 1818.37$ | 1.05 | $30.88 \pm 0.95$ | $9881.54 \pm 784.52$ | 0.02 |
| MLP (features, TD-0 Loss) | $12074.4 \pm 1819.26$ | 1.05 | $38.31 \pm 0.65$ | $9885.00 \pm 784.35$ | 0.02 |
| RNN (features, BCE Loss) | $14883.6 \pm 1516.95$ | 4.46 | $108.11 \pm 0.91$ | $12381.49 \pm 783.54$ | 0.09 |
| RNN (features, TD-0 Loss) | $14900.4 \pm 1517.40$ | 4.46 | $137.83 \pm 0.37$ | $12381.46 \pm 783.42$ | 0.12 |
| RNN (top-10, BCE Loss) | $13308.38 \pm 887.27$ | 0.48 | $72.67 \pm 0.53$ | $10296.98 \pm 382.87$ | 0.06 |
| RNN (top-10, TD-0 Loss) | $13506.19 \pm 887.69$ | 0.48 | $97.12 \pm 0.52$ | $10496.06 \pm 382.87$ | 0.08 |

*Table 15.* Computational cost of all failure prediction methods trained on $\pi_0$-FAST LIBERO-10 benchmark. Columns report reserved and peak GPU VRAM (MB), number of trainable parameters (M), total training wall-clock time (s), and per-epoch time (s), averaged over five seeds. Compared to OpenVLA (Table 14), all methods are substantially more efficient: peak VRAM stays below 3.2 GB and training completes in under 40 s. LSTM (TD-0 Loss) – same params as BCE uses the same architecture and hyperparameters as LSTM (BCE Loss), providing a controlled comparison of loss functions. TD-0 adds only ~13% to training time relative to BCE.

| run_name | vram reserved mb | num params M | train wall clock sec | peak vram mb | epoch time sec |
|---|---|---|---|---|---|
| MLP (BCE Loss) | $2320.00 \pm 411.44$ | 0.52 | $17.99 \pm 0.56$ | $1834.79 \pm 174.62$ | 0.01 |
| MLP (TD-0 Loss) | $2366.00 \pm 411.44$ | 0.52 | $20.67 \pm 0.64$ | $1837.03 \pm 174.63$ | 0.01 |
| LSTM (BCE Loss) | $4101.60 \pm 533.08$ | 2.36 | $22.31 \pm 0.29$ | $3106.82 \pm 165.88$ | 0.01 |
| LSTM (TD-0 Loss) – same params as BCE | $4123.60 \pm 533.08$ | 2.36 | $25.34 \pm 0.48$ | $3128.65 \pm 158.33$ | 0.01 |
| LSTM (TD-0 Loss) | $2619.20 \pm 305.89$ | 2.36 | $39.79 \pm 0.82$ | $2095.10 \pm 174.91$ | 0.03 |
| TDQC (top-10 BCE Loss) | $2450.00 \pm 411.44$ | 1.25 | $32.09 \pm 0.41$ | $1868.36 \pm 174.61$ | 0.02 |
| TDQC (top-10 TD-0 Loss) | $2448.00 \pm 411.44$ | 1.25 | $37.90 \pm 0.33$ | $1906.37 \pm 174.44$ | 0.03 |

temporarily stuck while trying to drop the alphabet soup into the basket around step 140. Then decreases after recovery around step 275, and increases again when the robot attempts to pick up the tomato soup at step 336, since grasping can fail. Overall, these scores are intuitive.

Figure 8 compared BCE and TDQC-based methods. TDQC methods are more sensitive to changes in policy behavior and produce sharper, more localized responses over time. In contrast, BCE-based methods tend to vary more smoothly and exhibit less sensitivity to short-term policy changes. For MLP models, however, TD loss does not appear to improve prediction quality.

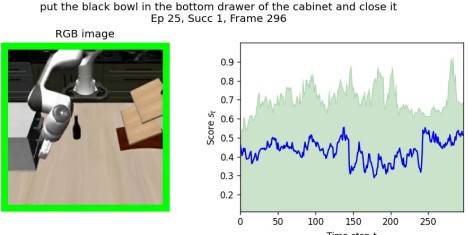 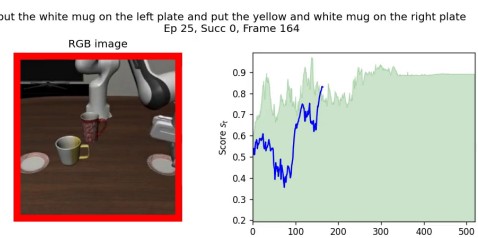

*(a)* Example of successful episode detected in RNN-TDQC (top-10 probabilities)

*(b)* Example of a failed episode detected by RNN-TDQC (top-10 probabilities) in the middle of the rollout

*Figure 6.* Failures and successes detected by RNN-TDQC (top-10 probabilities) align with the actual robot failures, as shown in the observations from OpenVLA + LIBERO-10 simulation. The green-shaded areas show the functional CP band. Once failure scores exceed the band, a failure flag is raised.

### G.6. WidowX analysis

The high ROC-AUC results in Table 2 on WidowX are obtained with SAFE-MLP, a heuristic that uses a cumulative loss, see Sec. 6.3, which appears to work well in terms of ROC-AUC in this specific benchmark. However, its connection with calibration is unclear. Note that SAFE-MLP is not reported in Table 1 since it cannot be used as a calibration metric. Figure 1

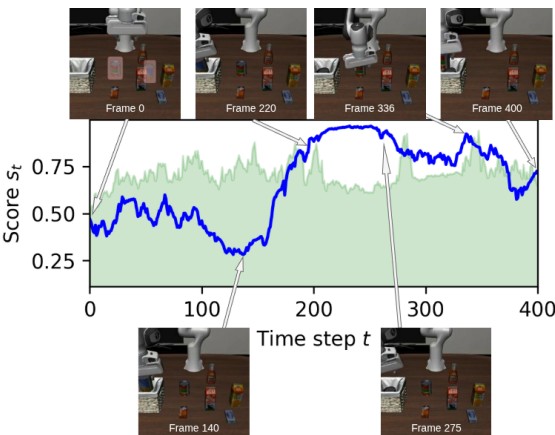

*Figure 7.* **Successful rollout with informative failure scores of TDQC top 10 probabilities on OpenVLA LIBERO-10 benckmark.** task: "put both the alphabet soup and the tomato sauce in the basket". The failure score rises when the policy becomes temporarily stuck while trying to drop the alphabet soup into the basket around step 140. Then, failure score decreases after recovery around step 275, and increases again when the robot is positioned himself and attempts to pick up the tomato soup at step 336.

shows that w.r.t. a standard MLP/LSTM, the action-based predictor is competitive on WidowX. Additional video examples are provided on our project webpage.

## G.7. Ablation Studies

We ablate several ways of incorporating temporal-difference learning when training a success predictor on OpenVLA in the LIBERO-10 benchmark. Specifically, we compare:

- **TD-0:** one-step bootstrapping with a standard MSE objective on the predicted success probability.

- **TD-$\lambda$:** multi-step bootstrapping using eligibility traces, which interpolates between TD-0 and Monte-Carlo targets via the trace parameter $\lambda$.

- **Categorical TD-0:** a distributional variant where the scalar success target is represented as a categorical vector and trained via Binary cross-entropy loss, following the setup of (Farebrother et al., 2024).

For TD-$\lambda$, we evaluate two trace values, $\lambda \in \{0.5, 0.8\}$, to study the effect of shorter versus longer credit assignment on sequential calibration.

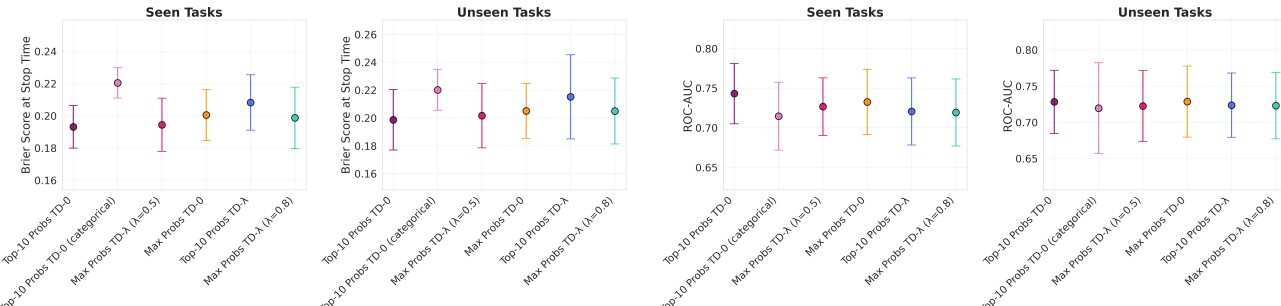

*(a)* Brier scores evaluated at the minimal task time for all ablation methods. We see here that TD-0 with top 10 probabilities achieves the lowest Brier score

*(b)* ROC-AUC scores evaluated at the minimal task time for all ablation methods. We see here that TD-0 with top 10 probabilities achieves the highest ROC-AUC with the lowest variance

*Figure 9.* **Ablation results for TD methods** Overall, we see that TD-0 with the top 10 probabilities achieve best performance

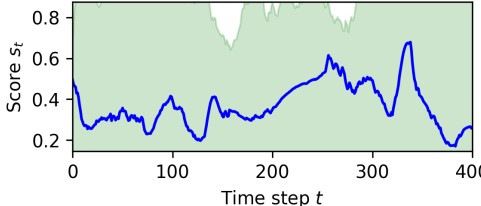

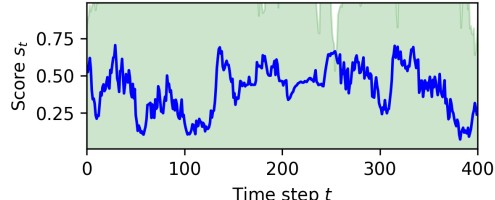

*(a)* Example of failure score of a successful episode detected in SAFE-RNN-BCE (features). This is the same rollout as in Figure 7.

*(b)* Example of failure score of a successful episode detected in SAFE-RNN-TDQC (features). This is the same rollout as in Figure 7.

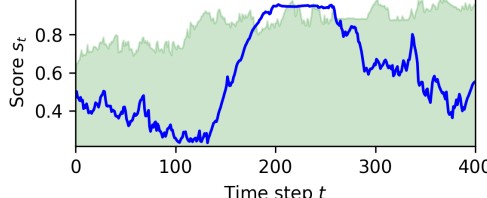

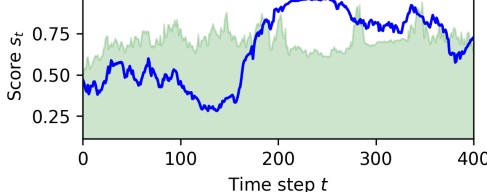

*(c)* Top-10 probabilities over time for RNN-BCE on the same successful rollout.

*(d)* Top-10 probabilities over time for RNN-TDQC on the same successful rollout.

*Figure 8.* Comparison between BCE and TD-based methods for the same rollout as in Figure 7. The top row shows the failure scores on features, while the bottom row shows the failure score on top-10 probabilities over time.

*Table 16.* Performance degregation as a function of failed trajectories, evaluated on $\pi_0$-FAST LIBERO-10 (unseen tasks). We vary the proportion of failed trajectories retained during training from 100% down to 10%. Performance degrades gracefully until 30% but drops more sharply at 10%, suggesting the method is robust to moderate class imbalance.

| Failed traj. % | Brier score ↓ | ROC-AUC ↑ |
|---|---|---|
| 100% | $0.097 \pm 0.033$ | $82.94 \pm 10.7$ |
| 60% | $0.101 \pm 0.040$ | $79.52 \pm 12.8$ |
| 30% | $0.100 \pm 0.045$ | $76.99 \pm 16.6$ |
| 10% | $0.135 \pm 0.065$ | $59.49 \pm 16.7$ |

### G.8. Sensitivity of the Early Stopping

Figure 10 reports an extended analysis of TPR and FPR across significance levels, averaged over 21 seeds. A steep ascent in the TPR curve indicates that even tight thresholds successfully catch most failures, reflecting confidently high failure scores. TD-based methods consistently outperform BCE-trained predictors in this regard. Since CP bands are calibrated on *successful rollouts* (negative examples only), the FPR is lower-bounded by the i.i.d. assumption, represented by $Y = X$ line in the plot. Deviations from this line reflect distribution shift between calibration and test tasks: a small shift produces results close to $Y = X$, while a larger shift, as seen in LIBERO, moves the curve further away. Notably, even under larger distributional shift, probability-based methods generalize better than feature-based methods.

### G.9. Degradation with fewer failed trajectories

We evaluated LSTM-TD0 on $\pi_0$-FAST LIBERO-10 with 100%, 60%, 30%, and 10% osf failed trajectories retained during training. Results on unseen tasks are shown in Table 3 in the attached link. The analysis shows that performance degrades gracefully until 30% and more sharply at 10%, suggesting robustness to moderate class imbalance.

### G.10. ROC-AUC vs Brier Score extended evaluation

We add experimental results across six benchmarks of the relationship between uncertainty calibration and ROC-AUC in several VLA models. Figures 11a through 11f illustrate a critical relationship in the evaluation of VLA models: the link between a model's **uncertainty calibration** and its **predictive performance**. All six scatter plots demonstrate a consistent

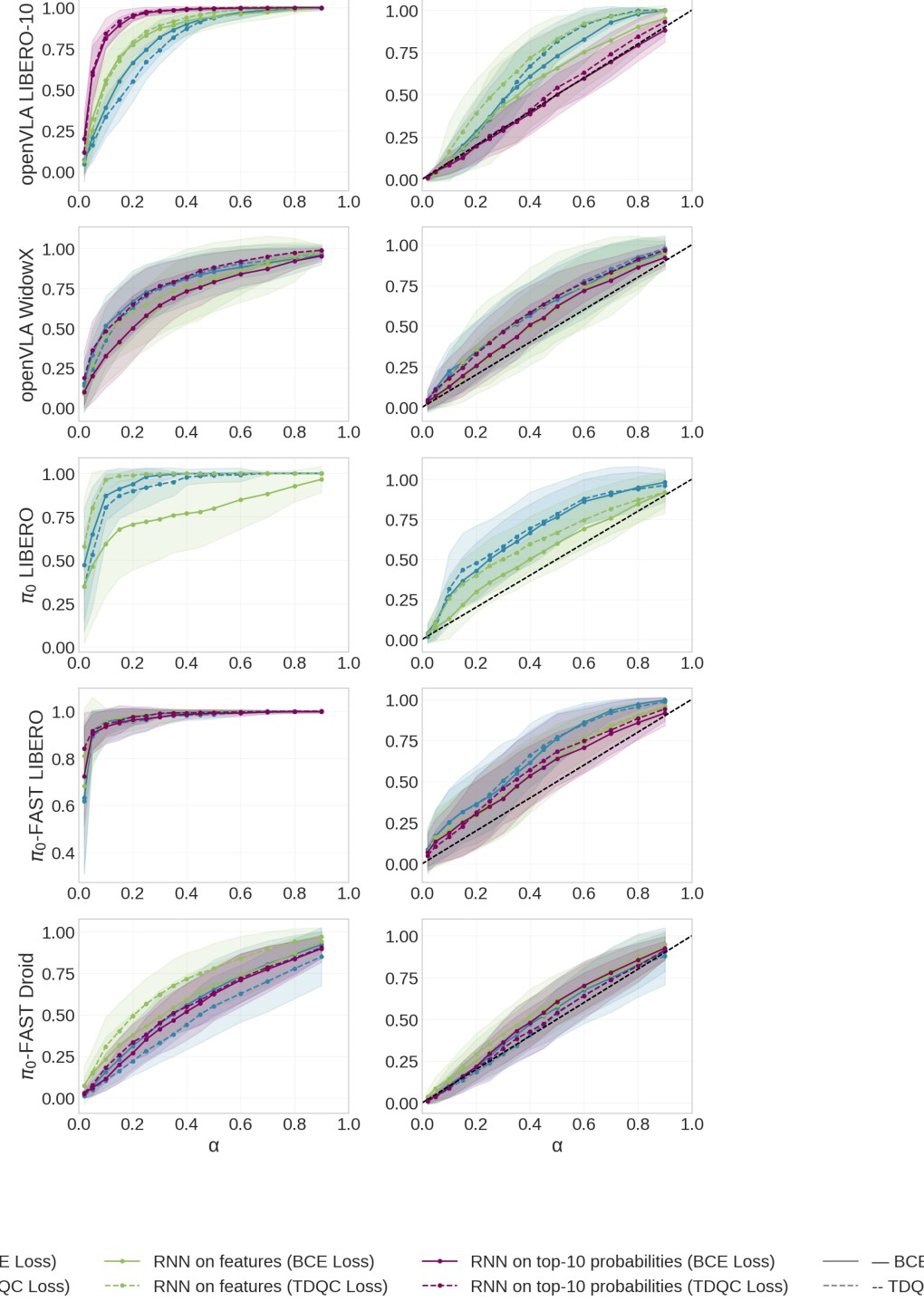

*Figure 10.* **Additional failure detection analysis using thresholds obtained by functional CP.** These plots show TPR (True positive rate, left column), and FPR (False positive rate, right column), w.r.t. the significance level $\alpha$, for each evaluation benchmark. These plots are averaged across 21 seeds.

and strong negative correlation, with Spearman's $\rho$ ranging from $-0.648$ to a peak of $-0.881$ for OpenVLA on WidowX. This indicates that as the Brier Score at Stop Time ($\hat{T}$) increases (representing higher calibration error), the ROC-AUC significantly decreases.

The Brier Score measures the accuracy of probabilistic predictions. These results confirm that **well-calibrated models** (lower Brier scores) are substantially more capable of distinguishing between successful and failing states, resulting in a higher ROC-AUC.

The lower correlation in UniVLA suggests that its internal uncertainty estimates are less reliable predictors of task success compared to the $\pi_0$ family and OpenVLA. In Figure 11c, UniVLA data points are more dispersed around the fit line, indicating that a low Brier score does not guarantee high ROC-AUC as consistently as it does for other models.

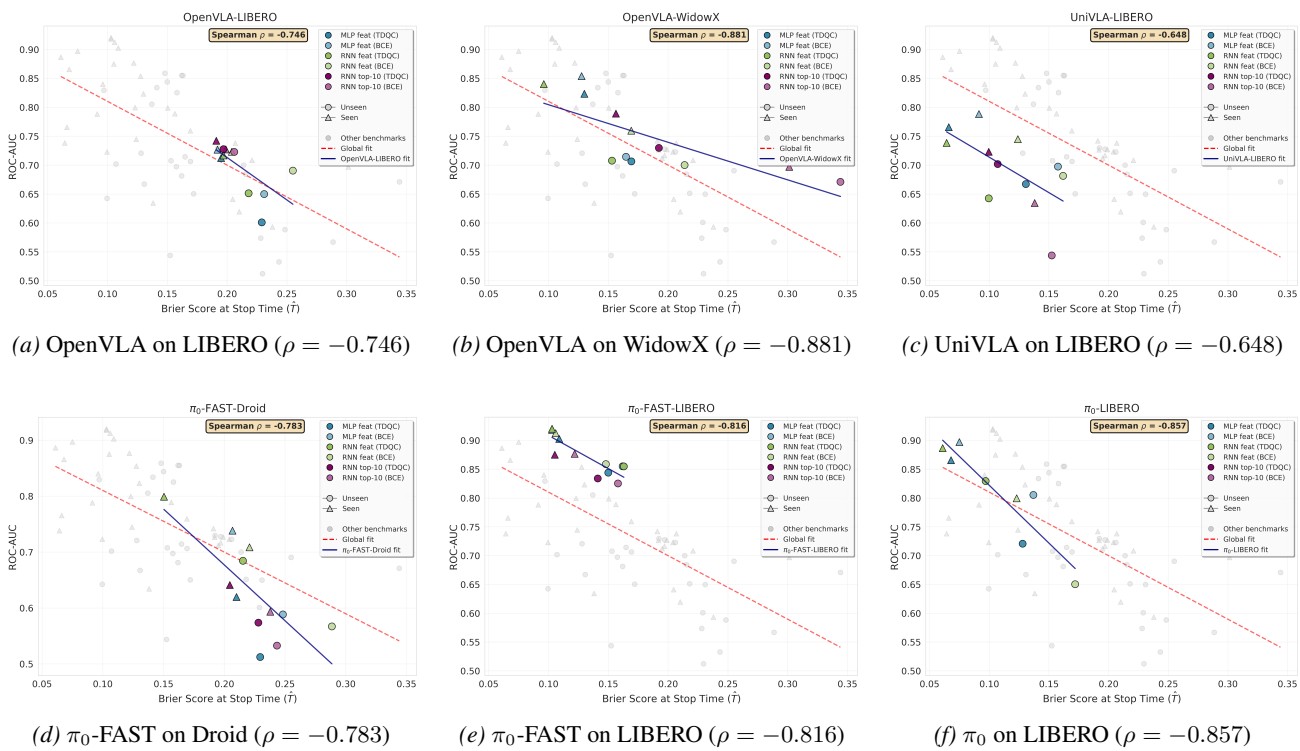

*(a)* OpenVLA on LIBERO ($\rho = -0.746$)   *(b)* OpenVLA on WidowX ($\rho = -0.881$)   *(c)* UniVLA on LIBERO ($\rho = -0.648$)

*(d)* $\pi_0$-FAST on Droid ($\rho = -0.783$)   *(e)* $\pi_0$-FAST on LIBERO ($\rho = -0.816$)   *(f)* $\pi_0$ on LIBERO ($\rho = -0.857$)

*Figure 11.* **Analysis of VLA Calibration and Success Rates.** (a-f) Scatter plots showing the strong negative correlation between Brier Score at Stop Time ($\hat{T}$) and ROC-AUC across different model-benchmark pairs.

### G.11. Relation between sequential Brier and ECE

Let us be reminded that the Brier score relates to the calibration and accuracy of the classifier by its two-component decomposition. Let $F = f(X)$ denote the random event that the model predicts a particular value, and let $\eta(F) = \mathbb{P}(Y = 1 \mid F)$, the success probability conditioned on the prediction. The Brier score can be decomposed as

$$\text{BS}(f) = \mathbb{E}\big[(F - Y)^2\big] = \mathbb{E}\big[(F - \eta(F))^2\big] + \mathbb{E}\big[\eta(F)(1 - \eta(F))\big].$$

The first term relates to *calibration*: the discrepancy between the predicted success probabilities and the true conditional event frequencies. The second term in the Brier score decomposition relates to accuracy, and measures how informative the score is about the label.

In order to prove this decomposition, we compared between the sequential Brier scores and ECE scores of all models and benchmarks, across 5 time quantiles (0.0, 0.2, 0.4, 0.6, 0.8) in Figure 16. Figures 12 to 15 shows the Sequential Brier score and ECE for seen and unseen sets respectively for different quantile times. In Figure 16 we report Spearman $\rho$ which measures monotonic relationship between variables, uses rank ordering rather than actual values and ranges between

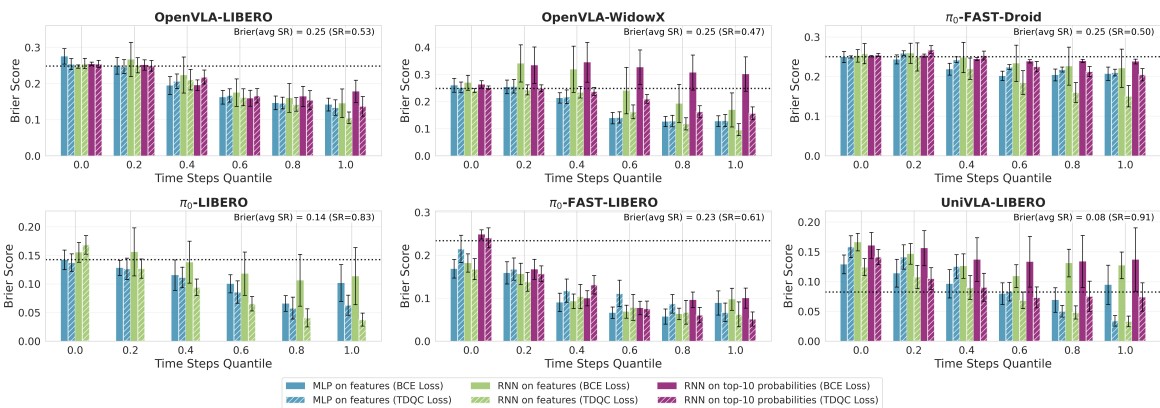

*Figure 12.* **Brier scores val-seen** sequential Brier score (lower is better) on the *seen* validation set averaged over 21 random seeds and all environments. We report Brier score in different *time quantiles*, where each subplot corresponds to a (model-benchmark) pair. For $\pi_0$, action probabilities are not directly interpretable, hence probability-based TDQC variants are not reported. Across all settings, our TD-based methods consistently outperform conventional predictors trained with binary cross entropy (BCE). For $\pi_0$ action probabilities are not directly interpretable, hence probability-based TDQC variants are not reported. The dotted horizontal line represents the Brier score of a constant predictor that consistently outputs the empirical mean success rate computed over the seen tasks.

$\rho \in [-1, 1]$. In OpenVLA and $\pi_0$ we see a strong monotonic relation between ECE and Brier scores where the red dashed line is the linear fitting on the points. We also note that $\pi_0$-FAST model does not show a strong relationship between the two metrics, which indicates that this model is less interpretable, and that the sequential Brier score affected more on the ROC-AUC component rather then the calibration component.

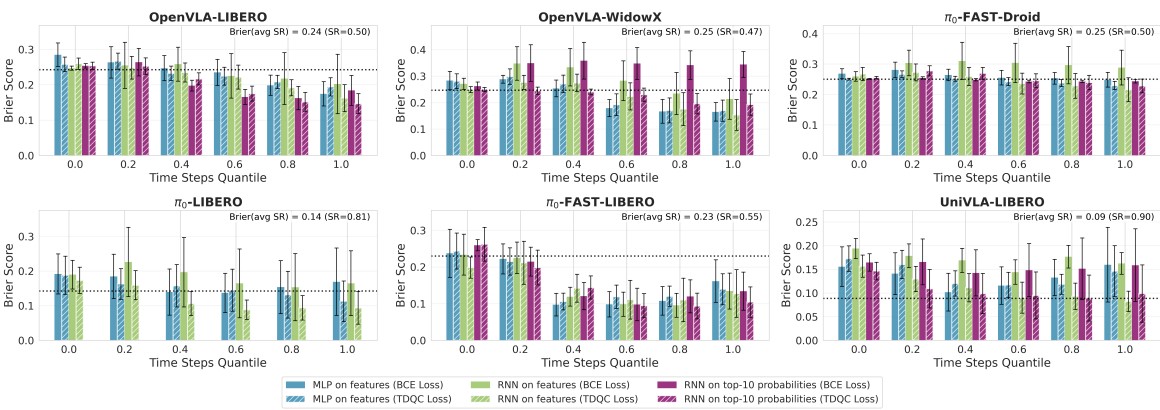

*Figure 13.* **Brier scores val-unseen** sequential Brier score (lower is better) on the *unseen* validation set averaged over 21 random seeds. We report Brier score in different *time quantiles*, where each subplot corresponds to a (model-benchmark) pair. For $\pi_0$, action probabilities are not directly interpretable, hence probability-based TDQC variants are not reported. Across all settings, our TD-based methods consistently outperform conventional predictors trained with binary cross entropy (BCE). For $\pi_0$ action probabilities are not directly interpretable, hence probability-based TDQC variants are not reported. The dotted horizontal line represents the Brier score of a constant predictor that consistently outputs the empirical mean success rate computed over the seen tasks.

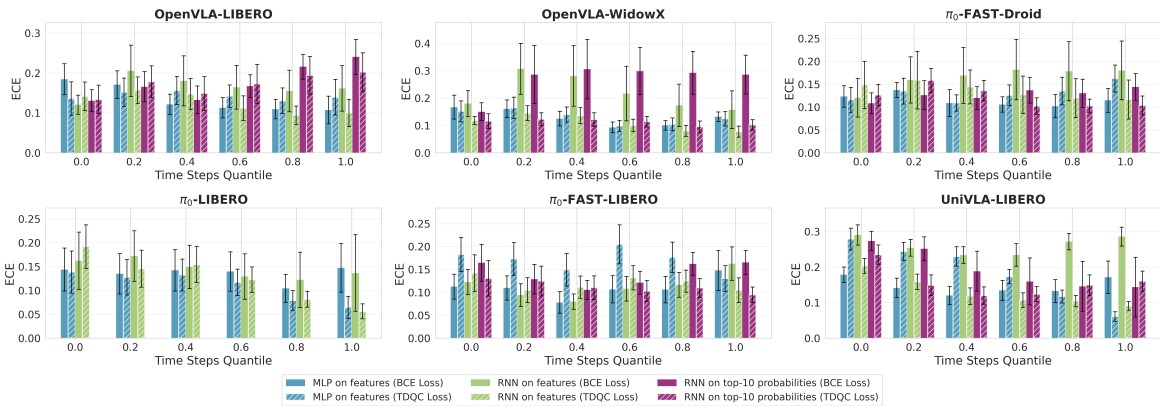

*Figure 14.* **ECE scores val-seen** ECE scores (lower is better) on the *seen* validation set averaged over 21 random seeds and all environments. We report ECE scores in different *time quantiles*, where each subplot corresponds to a (model-benchmark) pair. We see the correlation between lower Brier score and lower ECE scores in almost all settings. This highlights the Brier score decomposition shown in Section 2.1

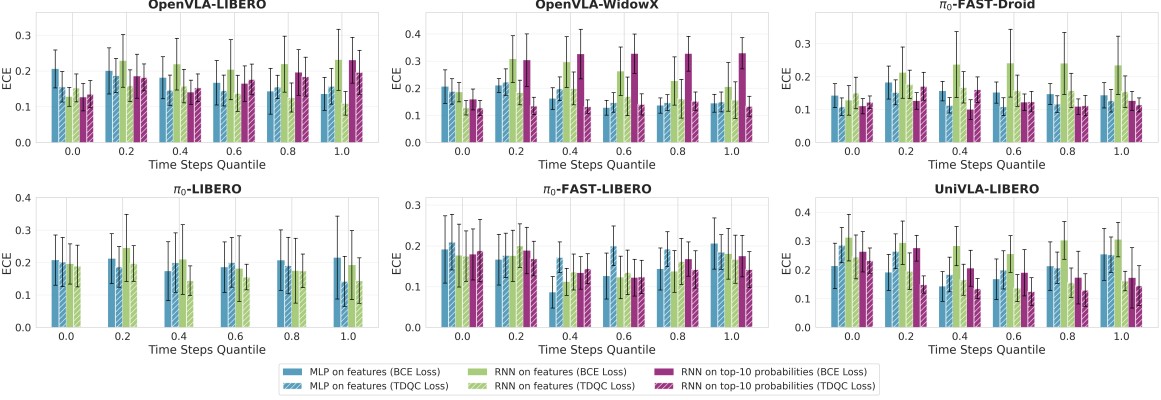

*Figure 15.* **ECE scores val-unseen** ECE scores (lower is better) on the *unseen* validation set averaged over 21 random seeds and all environments. We report ECE scores in different *time quantiles*, where each subplot corresponds to a (model-benchmark) pair. We see the correlation between lower Brier score and lower ECE scores in almost all settings. This highlights the Brier score decomposition shown in Section 2.1

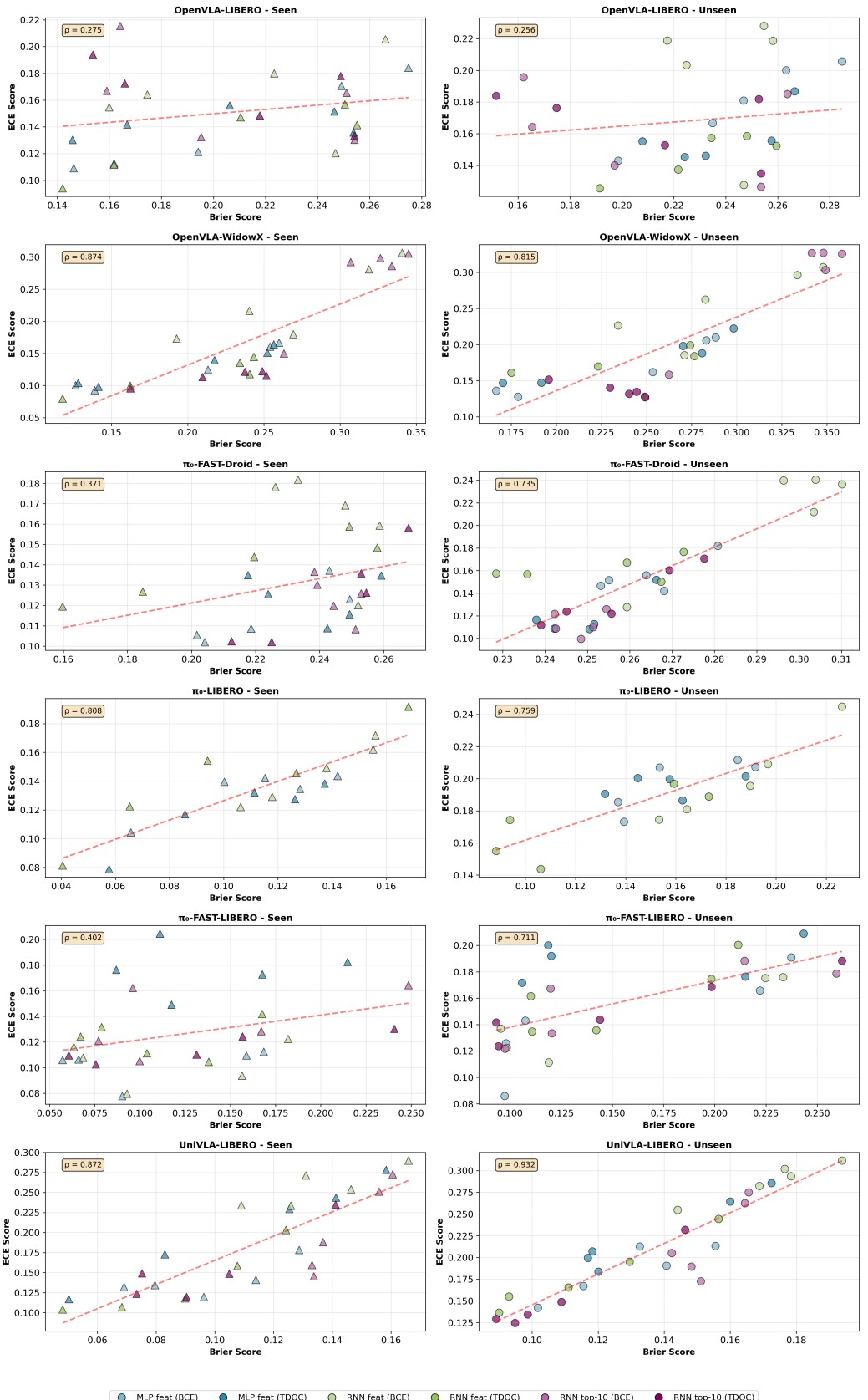

*Figure 16.* **ECE vs Brier scores** We compared ECE scores to sequential Brier scores in all models and benchmarks.

