# OpenReview forum: "Temporal Difference Calibration in Sequential Tasks: Application to Vision-Language-Action Models"
_ICML.cc/2026/Conference — ICML 2026 regular_

### Official Review · Reviewer_KvAp · 2026-02-28

**Soundness:** 3
**Presentation:** 3
**Significance:** 3
**Originality:** 3
**Overall Recommendation:** 5
**Confidence:** 5

**Summary:**

This paper addresses the problem of uncertainty quantification in sequential robotic tasks, specifically focusing on Vision-Language-Action (VLA) models. The authors argue that traditional calibration methods, which focus on single-step decision-making, fail to capture the cumulative nature of task success. They introduce a sequential extension of the Brier score and demonstrate that its risk minimizer corresponds to the value function of the VLA policy. Leveraging this connection, the authors propose TDQC (Temporal Difference Quantile Calibration), a method that uses TD-learning to train a calibration head on top of frozen VLA models. The method is evaluated on failure detection and early stopping tasks using simulated (LIBERO) and real-world (WidowX, Franka) datasets.

**Compliance With Llm Reviewing Policy:**

Affirmed.

**Final Justification:**

This is a solid paper, and the authors have addressed my concerns.

**Key Questions For Authors:**

- How does the performance of the TD-0 loss degrade as the proportion of failed trajectories in the training set decreases?
- Since the auxiliary predictor is essentially a value function, why is its application limited to early stopping? Could this predictor be used at inference time to rank multiple action candidates from the VLA, effectively using the "calibration" signal to improve the policy's success rate rather than just halting it? (As what has been done in [1])



[1] Steering Your Generalists: Improving Robotic Foundation Models via Value Guidance

**Limitations:**

The authors have not adequately discussed the limitations or potential negative societal impacts.

**Strengths And Weaknesses:**

Strengths:

- The paper provides a formal connection between uncertainty quantification and reinforcement learning. It proves that for binary outcomes, the risk minimizer of the sequential Brier score coincides with the policy's value function. Specifically, Theorem 5.1 establishes that minimizing the Mean Squared Error (MSE) of success prediction is equivalent to approximating the Q-function

- Unlike previous methods like SAFE that rely on Monte Carlo (MC) or cross-entropy losses, TDQC introduces Temporal Difference (TD) learning to calibration. By bootstrapping future value predictions to previous steps, TDQC achieves a superior bias-variance tradeoff compared to supervised training on terminal labels. The TD loss enforces consistency between consecutive time steps, leading to more stable and accurate success probability estimates as the trajectory progresses.

- A major practical advantage is the Policy-Restricted success predictor. TDQC can achieve state-of-the-art results using only action probabilities (tokens) as input. No weight access required is essential for proprietary models where internal hidden states are not accessible via APIs, allowing users to calibrate models without knowing the underlying architecture.

- The method consistently outperforms existing baselines across diverse benchmarks. TDQC achieved new state-of-the-art results in failure detection (ROC-AUC) for models like OpenVLA and $\pi_{0}$ on benchmarks including LIBERO and real-world Franka robot datasets. The high negative correlation between Brier Score and ROC-AUC suggests that better-calibrated probabilities directly translate to more reliable failure detection.

Weaknesses:

- It remains unclear if the TD-based calibration would remain stable or meaningful in non-Markovian, long-horizon, open-world tasks where "success" is defined by intricate temporal logic.

- The paper relies entirely on a binary definition of success. This is a significant regression compared to the state-of-the-art in robotic evaluation. As highlighted by frameworks like AutoEval, robust assessment of VLA models requires granular performance metrics, such as task progress, efficiency, or qualitative execution scores. A calibration head that only predicts the final binary outcome ignores the "partial success" states crucial for real-world safety and recovery.

- The use of TD-learning on offline robotic data—especially when paired with non-linear function approximators like the LSTMs used in this paper—is notoriously susceptible to divergence and overestimation (the "Deadly Triad") [1]. The authors provide no rigorous analysis of the convergence stability of the TDQC head across different data distributions or under distribution shift.

- The framework is strictly an offline policy evaluation tool. The method focuses on predicting failure rather than correcting the robot's behavior. It treats the VLA as a fixed entity and does not utilize the learned "value function" to improve the actual success rate of the task. The primary application is "early stopping" to avoid damage, which is a defensive safety mechanism rather than a method for improving decisions.

- The method requires a dedicated $\mathcal{D}_{cal}$ dataset containing both successful and failed trajectories generated by the target policy. In many real-world applications, generating and labeling failed trajectories can be expensive or dangerous. The predictor's accuracy is tied to the exchangeability of the calibration and test data; any significant shift in environment or task may require a new calibration set.

- While the method works well for OpenVLA, it struggles with models like $\pi_{0}$ and $\pi_{0}$-FAST. For such models, the system must fall back on internal hidden features, losing the "black-box" convenience.

- Each auxiliary predictor is often tuned or trained for specific task splits (Seen vs. Unseen), suggesting that a truly "universal" failure detector for generalist VLAs remains a challenge.

[1] Conservative Q-Learning for Offline Reinforcement Learning

---

> ### Author Rebuttal · Authors · 2026-03-31
>
> Thank you for your feedback! Additional figures and tables (unbalanced success rates, using the predictor for action ranking, TD applied to $\pi_0$-FAST without action probabilities) are reported here:
> https://anonymous.4open.science/r/TDQC_rebuttal-8EF2/README.md
> * **Q1: Degradation with fewer failed trajectories**: We evaluated LSTM-TD0 on $\pi_0$-FAST LIBERO-10 with 100%, 60%, 30%, and 10% of failed trajectories retained during training. Results on unseen tasks are shown in Table 3 in the attached link. The analysis shows that performance degrades gracefully until 30% and more sharply at 10%, suggesting robustness to moderate class imbalance. We also evaluated on UniVLA [1], which has a 10% failure rate, on LIBERO-10 and found that while TD still improves upon BCE, both methods underperform the uninformed baseline from Example 4.2.  (see Table 3 and Figure 6 in the link and response to Reviewer MtQF). For more analysis regarding distribution shift, see answer to R4qp and Fig. 4 in the link. Hence, improving calibration on extremely imbalanced datasets remains an important direction for future work.
> * **Q2: Using the predictor for action ranking**: We actually had the same idea and performed this experiment after the submission deadline! In addition to improving the success rate, we show that uncertainty estimation can be used to skip additional computations when success certainty is high. Our guided action-selection experiment uses the learned value function to score candidate actions via one-step look-ahead (using the simulator as a forward model), and chooses this action instead of the policy’s output. When the success rate is above a threshold, we skip the search and use the policy directly. We measured the success rates of OpenVLA on 3 unseen tasks from the LIBERO-10 benchmark. We compared the standard, unmodified OpenVLA policy (Baseline) against several value guided search configurations. The comparison is presented in Figure 5 in the attached link. It shows that TDQC methods outperform the baseline, reaching a 54% success rate, an improvement of 11% over the regular baseline. While the BCE loss variant also improves upon the baseline, it peaks lower at 51%. Further, we observe a near-linear tradeoff between additional compute and performance, using the success rate threshold. We note that this application requires look-ahead in a world model (here, simulator), and we will discuss it broadly in a new revision of the paper.
> * **W2: The paper relies entirely on a binary definition of success**: Note that our theoretical framework is not restricted to a binary definition of success (see Eq.2 in the paper). Our success notion generalizes to any threshold on the cumulative reward, and several thresholds can be considered simultaneously. In the reinforcement learning literature, this has been formulated as distributional value estimation [Bellmare et al 2017, A distributional perspective on reinforcement learning]. This formulation allows for prediction of the outcome distribution, opening the door to monitoring task progress and others. Extending and validating our approach in this setting is an important direction for future work.
> * **W6: Black-box is limited to OpenVLA**: While we agree that TDQC is not always applicable, as it requires the VLA output in the form of action probabilities, we realized it can be applied to many more models than what previously showed in the paper, by using the output layer logits. In Fig. 1, we report the new results for TDQC on $\pi_0$-FAST. This supports the applicability of the black-box approach to a broad set of models that do not output explicit action probabilities.
> * **W1 & W3: Convergence stability of TDQC**: We do not understand the concern. Deadly triad refers to off-policy training, while our training is on-policy. Evaluation on unseen tasks is done with a frozen network. We use standard TD training with a target function and did not encounter any convergence issues, which is in line with the RL literature since DQN [Mnih et al., 2015].
>
> [1] Bu, Qingwen, et al. "Univla: Learning to act anywhere with task-centric latent actions." arXiv preprint arXiv:2505.06111 (2025).

---

> > ### Author Rebuttal · Reviewer_KvAp · 2026-04-02
> >
> > My concerns have been addressed, and I raise the score to 5.

---

### Official Review · Reviewer_k7R1 · 2026-03-10

**Soundness:** 3
**Presentation:** 2
**Significance:** 3
**Originality:** 3
**Overall Recommendation:** 5
**Confidence:** 3

**Summary:**

The paper addresses sequential uncertainty quantification in VLA models by formalizing "sequential calibration" for episodic tasks. The authors introduce a sequential Brier score and prove that for binary outcomes, its risk minimizer is the policy’s value function. Based on this, they propose Temporal-Difference Q-based Calibration (TDQC), which uses TD learning to train a success predictor. Notably, TDQC can function as a policy-restricted predictor by using only output action probabilities, enabling calibration of black-box models. Evaluations across OpenVLA and π0 architectures show that TD-based objectives significantly improve failure detection and calibration over supervised baselines.

**Compliance With Llm Reviewing Policy:**

Affirmed.

**Final Justification:**

The rebuttal addressed my concerns. Thus I increased my score.

**Key Questions For Authors:**

- The action-probability-based TDQC underperforms hidden-dimension methods (like SAFE) in the WidowX environment. Given that action tokens are a highly compressed, discrete representation of the policy's intent, do you believe there is a fundamental information-theoretic limit to using token probabilities for calibration in complex manipulation tasks, or is this a failure of the predictorto reconstruct a sufficient state/value representation? I personally believe the former.
- The authors demonstrate that TDQC performs remarkably well when restricted to VLA-based features, such as action probabilities and hidden states. However, could the authors provide an upper-bound analysis by training the predictor on an observation-based history? This would serve as a performance bound for both TQDC and SAFE. While this might reduce the environment-agnostic nature of the value function, it would clarify how much predictive signal is lost when the success predictor is restricted to the policy's internal or external output space versus the full state/observation space.

**Limitations:**

yes

**Strengths And Weaknesses:**

Strengths
- The methodology is simple and technically sound. The use of TD learning for propagating failure signals is well motivated and integrates naturally with value learning formulations.
- Overall well-written paper

Weaknesses
- While the experiments suggest that TD learning improves performance, the paper lacks sufficient evidence or analysis to determine the actual limits of action-probability-based failure prediction. Although the authors do not claim the problem is solved, it would be nice to see a deeper investigation into the current limitations of this approach. Specifically, the paper provides no error analysis or even hypothesis to explain why the action-based predictor underperforms on the WidowX benchmark compared to other environments.
- As written, the formulation can resemble training a standard value function from observations, which obscures that the goal is to estimate the value function using model-based features rather than relying on environment-specific value estimates.
- Important implementation details are difficult to locate. In particular, the predictor architecture and related implementation details are relegated to the appendix. Even a short description in the main paper would improve clarity.
- The novelty is somewhat overstated. Conceptually, the approach can be interpreted as training a value estimator that predicts failure events. While the framing around calibration and action probability prediction is interesting, the underlying idea is exctremely closely related to standard value estimation.

---

> ### Author Rebuttal · Authors · 2026-03-31
>
> Thank you for your feedback! Additional figures and tables are reported here: https://anonymous.4open.science/r/TDQC_rebuttal-8EF2/README.md
> We will add a brief description of the implementation in the main text.
> * **Q1: information-theoretic limit**: Token probabilities are functions of the history. We believe the info-theoretic limit is the mutual information between the history and the success/failure event. In principle, since taking actions only depends on the argmax of action probabilities, there is room to convey more information in the action probabilities without changing the policy performance. However, the actual information content depends on the policy training, which we do not control. We currently do not know how to relate this information content to, e.g., max-likelihood training loss. It's an interesting question for future work!
> * **W1: WidowX performance**: The high ROC-AUC results in Table 2 on WidowX are obtained with SAFE-MLP, a heuristic that uses a cumulative loss, see Sec. 6.3 lines 378-390, which appears to work well in terms of ROC-AUC in this specific benchmark. However,  its connection with calibration is unclear, and note that SAFE-MLP is not reported in Table 1 since it cannot be used as a calibration metric. Note from Figure 1 (see link) that w.r.t. a standard MLP/LSTM, the action-based predictor is competitive on WidowX.
> * **Q2: upper bound via observation-based training**: The issue is that it is very easy to overfit with observation input. Our training data (Libero) has 7 tasks with 50 trajectories each, and we test on unseen tasks. To validate this, we trained an LSTM with raw images (processed via ResNet-18/50) as inputs on OpenVLA-LIBERO 10. Recall that the hidden-state LSTM achieves ROC-AUC 72.30 (seen) / 69.04 (unseen) and sequential Brier score 0.204 / 0.255. The image-based LSTM achieves ROC-AUC 65.68 (seen) / 48.60 (unseen) and Brier score 0.288 / 0.413.
> * **W4: The novelty is somewhat overstated**: We agree with the reviewer that our approach is closely related with value estimation. However, we note that the link between sequential calibration and value estimation is uncovered by our framing and theory and, to the best of our knowledge, wasn’t known before. This opens the door to a variety of value estimation methods for calibration in sequential tasks. Thus, we see bridging together value estimation and calibration as a main contribution of our work rather than a weakness.

---

> > ### Author Rebuttal · Reviewer_k7R1 · 2026-04-03
> >
> > My concerns are resolved thanks to the response of the authors. I will increase my score accordingly.

---

### Official Review · Reviewer_R4qp · 2026-03-13

**Soundness:** 3
**Presentation:** 3
**Significance:** 3
**Originality:** 3
**Overall Recommendation:** 4
**Confidence:** 4

**Summary:**

This paper focuses on the uncertainty calibration of Vision-Language-Action (VLA) models in sequential robotic tasks. It addresses a critical pain point where traditional calibration methods, designed for single-step decision-making, fail to account for delayed success feedback and the inter-dependencies between sequential steps. The authors provide the first formal definition of a calibration framework for sequential tasks and systematically prove a theoretical connection between minimizing the sequential Brier score and value function estimation in reinforcement learning.
Building on this theoretical foundation, the paper proposes TDQC (Temporal Difference Q-based Calibration), a sequential calibration method that achieves uncertainty estimation and failure detection solely through the single-step action probabilities of VLA models. This approach overcomes the limitations of existing methods that rely on accessing the model's internal hidden states. Extensive experiments conducted on the LIBERO simulation benchmark, as well as WidowX and Franka real-world robot datasets, demonstrate that TDQC outperforms state-of-the-art methods like SAFE in terms of sequential Brier scores and ROC-AUC for failure detection. Furthermore, the results show that the integration of TD loss significantly enhances calibration performance, proving that post-TD-calibrated action probabilities can effectively predict long-term task success.

**Compliance With Llm Reviewing Policy:**

Affirmed.

**Final Justification:**

This paper introduces a sequential Brier score formulation for episodic tasks and connects uncertainty calibration with RL value estimation, enabling TD learning to improve calibration over time.

While the rebuttal adequately addresses my concerns and strengthens the empirical analysis, I believe maintaining my original score remains the most appropriate overall assessment.

**Key Questions For Authors:**

(1) Quantitative Metrics on Computational Cost: Could the authors provide specific quantitative metrics for TDQC across various benchmarks, including total training duration, per-iteration time, and peak VRAM usage on the calibration datasets? Furthermore, please provide a direct comparison of training speeds between TDQC, SAFE, and other baselines under identical hardware and dataset conditions to clarify the computational overhead introduced by the step-by-step training approach.
(2) Training Efficiency and Batching: Regarding the sequential training nature of TDQC, have the authors explored any batch-processing schemes to improve training efficiency? If so, what are the trade-offs between performance and efficiency after such improvements? If not, what are the core technical bottlenecks that prevent effective batching for this method?
(3) Sensitivity Analysis of the Early Stopping Mechanism: The paper proposes an early stopping mechanism based on Conformal Prediction but only validates its overall effectiveness. Could the authors supplement this with quantitative results showing how different significance levels impact both stopping precision and task efficiency?
(4) Extensibility to LLMs: When extending TDQC to sequential calibration in Large Language Models (LLMs), how should "action probabilities" and "task success labels" be formally defined in the context of text generation? Additionally, would the sequential Brier score require specific modifications to adapt to the discrete and high-dimensional nature of linguistic sequences?

**Limitations:**

(1) Dependency on Calibration Data Quality and Distribution: The performance of TDQC heavily relies on a pre-collected offline calibration dataset. While the paper demonstrates some generalization to unseen tasks, it lacks a systematic stress test or robustness analysis under extreme Out-of-Distribution (OOD) scenarios, such as significant changes in environmental noise or lighting conditions. This raises concerns about the reliability of the TD predictor when deployed in unpredictable real-world environments.
(2) Lack of Computational and Scalability Analysis: The paper provides insufficient data regarding the additional computational overhead, memory footprint, and scalability of the TD step-by-step updates, especially when dealing with ultra-large-scale datasets. For resource-constrained, real-time embedded robotic systems, the absence of these metrics makes it difficult to evaluate the method’s actual feasibility for on-device deployment.
(3) Restriction to Binary Outcomes: The current framework is primarily designed for binary success/failure outcomes. Its extensibility to tasks involving continuous rewards or intermediate state scores remains unverified. This limitation may restrict the applicability of the proposed method in more complex robotic scenarios where task progress is non-binary or requires fine-grained evaluation.

**Strengths And Weaknesses:**

Strengths
(1) Solid Theoretical Foundation: The paper formally defines the calibration problem for sequential tasks and derives a decomposition of the sequential Brier score. It establishes a theoretical equivalence between minimizing this score and value function estimation in Reinforcement Learning (RL), proving that minimizing the sequential Brier score is equivalent to learning a calibrated task-success predictor.
(2) Algorithmic Innovation: The proposed TDQC method innovatively introduces the Temporal Difference (TD) loss from RL into sequential calibration. A key advantage is its "black-box" nature: it achieves high-performance calibration and low-cost prediction using only action probabilities, without requiring access to the internal hidden states of VLA models.
(3) Extensive and Convincing Evaluation: The authors conduct comprehensive experiments across multiple state-of-the-art VLA models (e.g., OpenVLA) and both simulation and real-world robotic benchmarks. By comparing against static baselines and SOTA methods like SAFE, the paper validates the effectiveness of TDQC through various metrics such as sequential Brier score and ROC-AUC. Furthermore, ablation studies effectively demonstrate the optimality of the TD-0 loss.
(4) Broad Generalizability: The proposed framework is potentially extensible to other sequential generation tasks, such as Large Language Models (LLMs). This provides a promising new perspective for addressing uncertainty calibration in LLM hallucination detection.

Weakness
(1) Lack of Analysis on Training Efficiency and Computational Cost: The proposed TDQC method employs a step-by-step training approach for trajectory data, whereas baselines like SAFE utilize batch training. However, the paper does not provide quantitative data regarding the training duration of TDQC on the calibration datasets. Furthermore, there is no direct comparison of training speed or computational overhead (such as peak VRAM usage or convergence rate) between TDQC and SAFE or other baselines. This omission makes it difficult to assess the scalability and practicality of the method when applied to large-scale trajectory datasets.
(2) Insufficient Evaluation of the Early Stopping Mechanism: While early stopping based on Conformal Prediction is highlighted as a key application, the experimental validation remains somewhat superficial. Specifically, the paper lacks a sensitivity analysis on the significance level , which is critical for balancing stopping precision and task efficiency. Additionally, the proposed mechanism is not compared against other standard early stopping strategies (e.g., simple thresholding on action entropy or value-based heuristics). The lack of in-depth analysis regarding hyperparameter tuning and comparative effectiveness leaves the robustness of this mechanism insufficiently justified.

---

> ### Author Rebuttal · Authors · 2026-03-31
>
> Thank you for your feedback! Additional figures and tables (training costs, early stopping) are reported here: https://anonymous.4open.science/r/TDQC_rebuttal-8EF2/README.md
>
> * **W1 & Q1: Computational cost**: For real robot deployment, test-time compute for TDQC is not more costly than baselines such as SAFE: both require only a forward pass of the MLP/LSTM at every time step, typically much lighter than the VLA forward pass. Furthermore, the network for TDQC with action probabilities as input is lightweight, and smaller than SAFE with features as input. That said, we agree with the reviewer that training costs are important to report.
> RL theory suggests that TD-based methods require longer training than Monte-Carlo (MC) methods. With linear function approximation, MC is equivalent to linear least squares, while TD is an iterative linear least squares procedure that converges exponentially fast [see Chapter 12.3.7 Projected Value Iteration in Mannor et al., 2025, Reinforcement Learning: Foundations]. With non-linear neural networks, both methods are trained with minibatch SGD, and we are not aware of a clear convergence rate result. We instead report empirical results. We evaluated all methods on the same GPU (NVIDIA RTX 5090) with 5 seeds on various computational metrics. Results are reported in Table 1 and 2 in the attached link. From our analysis, the overhead from the TD loss is small, ~25% on OpenVLA and ~15% on π₀-FAST:
>     * OpenVLA-LIBERO 10: MLP-BCE trains in 30.9 s (9.8 GB peak VRAM); MLP-TD0: 38.3 s (9.8 GB); LSTM-BCE: 108.1 s (12.3 GB); LSTM-TD0: 137.8 s (12.3 GB); TDQC-BCE: 72.67 s (10.2 GB); TDQC-TD0: 97.12 s(10.4GB).
>     *  $\pi_0$-FAST LIBERO 10, all methods train in under 40 s with <2.4 M parameters and <3.1 GB peak VRAM.
> * **Q2: Training efficiency and batching**: See our response above. We use minibatch SGD for all methods, which is standard for TD (e.g., the DQN paper Mnih et al., 2015). Table 2 in the link shows that, when using the same architecture and hyper parameters, TD training time for 1000 epochs was only 13% longer than BCE (3 seconds more).
> * **W2 & Q3: Sensitivity of the Early Stopping**: Fig. 4 (in the link) reports an extended analysis  of TPR and FPR across significance levels $\alpha$, averaged over 21 seeds. A steep ascent in the TPR curve indicates that even tight thresholds successfully catch most failures, reflecting confidently high failure scores. TD-based methods consistently outperform BCE-trained predictors in this regard. Since CP bands are calibrated on successful rollouts (negative examples only), the FPR is lower-bounded by $\alpha$ under the i.i.d. assumption, represented by Y=X line in the plot. Deviations from this line reflect distribution shift between calibration and test tasks: a small shift produces results close to Y=X, while a larger shift, as seen in LIBERO, moves the curve further away. Notably, even under larger distributional shift, probability-based methods generalize better than feature-based methods. For additional context, Tables 1 and 2 in the paper include comparisons against entropy-based thresholding baselines, and the updated Figure 1 adds a dotted-line baseline predicting success at the mean training-task success rate (see also our response to Reviewer MtQF).
> * **Q4: LLM extension**: The natural analogue of action probabilities is the LLM's next-token probability $p(T\_i | T\_{i-1})$. Task success can be defined at the sentence/answer level via downstream metrics (e.g., correctness verifiers or process reward models [Lightman et al., 2023]). The sequential Brier score requires no fundamental modification: it remains a proper scoring rule over predicted success probabilities conditioned on a partial token sequence. We note that extending TDQC to high-dimensional linguistic sequences is a promising direction for future work.
> * **L1: Limitation, Extreme OOD**: We agree, this is an important question that is currently not answered by our work (or any other work, to the best of our knowledge). We do believe that our contribution may help pave the way to such answers.
> * **L3: Limitation, Binary outcomes**: Note that our theoretical framework is not restricted to binary outcomes (see Eq. 2 in the paper). It generalizes to any threshold on the cumulative reward, and several thresholds can be considered simultaneously. In the reinforcement learning literature, this has been formulated as distributional value estimation [Bellmare et al 2017, A distributional perspective on reinforcement learning]. This formulation allows for prediction of the outcome distribution, opening the door to monitoring task progress and others. Extending and validating our approach in this setting is an important direction for future work. See answer to reviewer MtQF for all limitations.

---

> > ### Author Rebuttal · Reviewer_R4qp · 2026-04-02
> >
> > My concerns have been adequately addressed.

---

> > > ### Author Response · Authors · 2026-04-04
> > >
> > > We are grateful that our rebuttal addressed your concerns. All of the reviewers are reporting their concerned as "fully resolved". In view of this, does your current score still reflect your overall assessment of the paper?

---

### Official Review · Reviewer_MtQF · 2026-03-14

**Soundness:** 3
**Presentation:** 4
**Significance:** 3
**Originality:** 3
**Overall Recommendation:** 5
**Confidence:** 4

**Summary:**

The paper tackles the uncertainty quantification problem in sequential decision-making tasks. To address this, the authors introduce "sequential Brier score" for episodic tasks and prove that the predictor minimizing this sequential Brier score is the policy's value function. This result connects uncertainty calibration to Reinforcement Learning (RL), allowing the authors to apply Temporal-Difference (TD) learning to calibrate success predictors over time.

**Compliance With Llm Reviewing Policy:**

Affirmed.

**Final Justification:**

I believe the theoretical contribution is valuable and deserves to be published, and I appreciate the authors’ thorough rebuttals. Therefore, I am increasing my score and recommend acceptance.

**Key Questions For Authors:**

see above

**Limitations:**

please explicitly discuss limitations of the work

**Strengths And Weaknesses:**

Strengths

- The paper is very well-organized and structured. The need for calibration frameworks in sequential tasks is well motivated and the problem statement is clear. It was pleasant reading the paper. Material in the appendix is also appropriate and helpful in understanding details of the paper.

- The paper builds on a strong theoretical foundation. The definition of the sequential Brier score and its linking to RL value estimation provide a rigorous justification for using TD loss for calibration.

- The empirical results show that TD bootstrapping improves the calibration performance.

Weaknesses

- Limited evaluation: The analysis would greatly benefit from more diverse asepects, not just relying on quantitative analysis of Brier scores and ROC-AUC. For instance, providing qualitative or quantitative analyses of the model’s behavior during specific successful vs failed rollouts would offer a deeper understanding of how the method operates in practice.

- Incremental modification beyond SAFE: the method mainly replaces a MC loss with a TD loss. While effective, TD learning is a standard technique well-known to work well in the RL community. Furthermore, the standalone TDQC is not applicable to all benchmarks.

- The visual presentation of results (Figure 1) initially makes it appear as TDQC does not work well. The authors should improve Figure 1, particularly the label, and refine how it is referenced within the contribution statement on page 2.

- There is ambiguity in h_t and h_T. In particular, when defining Y(h_t) in eq (2), it is confusing whether the success condition should consider R(h_t) or R(h_T). It is currently defined in terms of R(h_t).

- The takeways of Example 4.2 is unclear. It would strengthen the paper if the authors could connect this intuition to the experiments.

- There are several formatting typos. In Section 2.2, P, O, R inconsistently alternate between standard fronts and calligraphic fonts. In the caption of Table 2, “-“ indicates that ROC-AUC can’t be calculated, not Brier score.

---

> ### Author Rebuttal · Authors · 2026-03-31
>
> Thank you for your feedback! Additional figures (revised Fig. 1, Example 4.2 baseline, qualitative analysis, TD applied to $\pi_0$-FAST without action probabilities) are reported here: https://anonymous.4open.science/r/TDQC_rebuttal-8EF2/README.md
> * **W1: Limited evaluation**: Quantitative Brier score and ROC-AUC analysis is essential for comparison with prior work. However, we agree that qualitative analysis is important. We gathered several insights from inspecting videos and failure modes:
>     * All methods reliably detect failure modes in which the policy gets stuck or oscillates with slow back-and-forth motions.
>     * For LSTM and TDQC methods, once the robot successfully grasps an object, the failure score drops, suggesting that the model recognizes this as a 'good' action.
>     * In the attached link, Fig. 2 shows a successful rollout with an informative failure score. The failure score rises when the policy becomes temporarily stuck while trying to drop the alphabet soup into the basket around step 140. Then decreases after recovery around step 275, and increases again when the robot attempts to pick up the tomato soup at step 336, since grasping can fail. Overall, these scores are intuitive.
>     * Compared with BCE, we found that TD-based methods are more sensitive to changes in policy behavior and produce sharper, more localized responses over time. In contrast, BCE-based methods tend to vary more smoothly and exhibit less sensitivity to short-term policy changes, see Fig. 3a-d for examples. We will expand this analysis in the revised version.
> * **W2: Incremental beyond SAFE**: Our main contribution is to formalize the problem and to link it to RL. This is essential to support using the TD loss, which has not been used before in this context. While our presentation of TDQC requires the VLA action probabilities, we realized it can be applied more broadly. Many VLAs use FAST tokenizer, which decodes tokens into discrete actions and exposes logits over the decoder vocabulary; We applied TDQC to $\pi_0$-FAST using these logits as input, and a TD loss, see Fig. 1 in the link. TDQC improves on the baselines. As an additional application of calibration beyond SAFE, see our response to Reviewer KvAp.
> * **W3 & W5: Figure1 and Example 4.2**: Following the reviewer suggestion, we have revised Fig. 1 of the paper with clearer labels and an additional static baseline based on Example 4.2. The new version (Fig. 1 in the attached link) clearly shows TD-based methods consistently outperforming BCE across all benchmarks and time quantiles. To connect Example 4.2 to the experiments, we plot the Brier score of a constant predictor outputting the empirical mean success rate on training tasks (the worst-case uninformative baseline, $\mathbb{E}\_\pi [ Y (h\_T)]$ in the example). TD methods consistently outperform this baseline, with the gap growing as more trajectory information becomes available. Interestingly, this constant baseline (which was not reported in prior studies) outperforms some BCE-based methods, such as LSTM on $\pi_0$-FAST Droid. We thus believe it is important as a minimal proof of information content in the trajectory.
> * **W4: Notation $h_t$ and $h_T$**: to avoid any confusion, we follow the reviewer’s suggestion and change $h_t$ into $h_T$ when defining a success.
> * **W5: Limitations**: We will add an explicit Limitations section covering: (1) limited cross-environment and cross-embodiment generalization; (2) restriction to binary episodic success in experiments (extension to dense rewards is future work); and (3) alternative application to calibration. The guided action-selection experiment requires a forward model, a gap that recent world models may help close [Gemini Robotics Team et al., 2026; Badithela et al., 2025]. See reviewer KvAp’s response 2 for details on the that experiment.

---

> > ### Author Rebuttal · Reviewer_MtQF · 2026-04-03
> >
> > I appreciate the additional experimental results.

---

> > > ### Author Response · Authors · 2026-04-04
> > >
> > > We are grateful that our rebuttal addressed your concerns. All of the reviewers are reporting their concerned as "fully resolved". In view of this, does your current score still reflect your overall assessment of the paper?

---

### Decision · Program_Chairs · 2026-04-30

**Decision:**

Accept (regular)

**Comment:**

The paper introduces a sequential calibration framework for episodic VLA tasks via a sequential Brier score, and shows (for binary outcomes) that the optimal calibrated predictor corresponds to the policy value function. Then they introduce a practical algorithm called TD-based calibration (TDQC).

Reviewers find the theory clean and the approach practically valuable, with experiments on LIBERO and real-robot data showing consistent gains in calibration/failure detection over prior methods, including a useful “black-box” variant using only action probabilities/logits.